

# Towards a more detailed representation of high-latitude vegetation in the global land surface model ORCHIDEE (ORC-HL-VEGv1.0)

Arsène Druel[1,2], Philippe Peylin[1], Gerhard Krinner[2,] Philippe Ciais[1], Nicolas Viovy[1], Anna Peregon[1,3], Vladislav Bastrikov[1], Natalya Kosykh[3] and Nina Mironycheva-Tokareva[3]

[1]Laboratoire des Sciences du Climat et de l'Environnement, CEA-CNRS-UVSQ CE Orme des Merisiers, 91 190 Gif sur Yvette, France
[2]CNRS, Univ. Grenoble Alpes, Institut des Géosciences de l'Environnement (IGE), F-38000 Grenoble, France
[3]Institute of Soil Science and Agrochemistry, Siberian Branch Russian Academy of Sciences (SB RAS), Novosibirsk, 630090, Ak. Lavrentieva ave., 8/2, Russia

*Correspondence to*: Arsène Druel (arsene.druel@gmail.com)

Journal: Geoscientific Model Development

**Abstract.** To improve the simulation of vegetation-climate feedbacks in the high latitudes, three new circumpolar Plant Functional Types (PFTs) were added in the ORCHIDEE land surface model, namely non-vascular plants (NVPs) representing bryophytes and lichens, arctic shrubs, and arctic C3 grasses. Non-vascular plants are assigned no stomatal conductance, very shallow roots, and can desiccate during dry episodes and become active again during wet periods, which gives them a larger phenological plasticity compared to grasses and shrubs. Shrubs have a specific carbon allocation scheme, and differ from trees by their larger survival rates in winter, due to protection by snow. Arctic C3 grasses have the same equations than in the original ORCHIDEE version, but different parameter values, optimized from in-situ observations of biomass and NPP in Siberia. In situ observations of living biomass and productivity from Siberia were used to calibrate the parameters of the new PFTs using a Bayesian optimization procedure. With the new PFTs, we obtain a lower Net Primary Productivity (NPP) by 31% (from 55°N), as well as a lower roughness length (-41%), transpiration (+33%) and a higher winter albedo (by 3.6%) due to a larger snow cover. A simulation of the water balance and runoff and drainage in the high northern latitudes using the new PFTs results in an increase of fresh water discharge in the Arctic ocean by 11% (+140 km$^{-3}$ y$^{-1}$), owing to less evapotranspiration. Future developments should focus on the competition between these three PFTs and boreal trees PFTs, in order to simulate their area changes in response to climate change, and the effect of carbon-nitrogen interactions.

## 1    Introduction

To understand the role of vegetation feedbacks in climate change, global land surface models included in Earth System Models (ESM) describe the carbon, water and energy exchanges between the vegetation and the atmosphere at large scales. To this end, Surface-Vegetation-Atmosphere transfer schemes (SVATs, e.g. Henderson-Sellers et al., 1996) were developed and coupled with General Circulation Models (GCMs) that provide the meteorological forcing used as input to SVATs. Several studies show that the terrestrial biosphere plays an important role in controlling the spatial and temporal distribution of carbon, water and energy fluxes, and thus indirectly in modulating regional to continental scale climate. Specifically, it appears that high-



latitude ecosystems have a significant impact on the climate (Bonan, 1995; Christensen et al., 1999; Chapin et al., 2000). For example, circumpolar vegetation changes have played an important role in the last glacial inception, i.e. 26.5 ka to 20 ka (Clark et al., 2009). Reduced tree cover led to an increase in albedo and snow cover, a reduction in temperature and precipitation and ultimately changes in atmospheric circulation and cooling of the high latitudes (Gallimore and Kutzbach, 1996; de Noblet et al., 1996; Meissner et al., 2003; Vavrus et al., 2008; Colleoni et al., 2009). In the circumpolar regions, critical physical processes are the dynamics of permafrost (Lawrence and Slater, 2005; Koven et al., 2011), snow deposition and cover and its effect on surface albedo, soil thermal dynamics and the impact of vegetation roughness length on momentum and flux exchanges with the atmosphere. While the Net Primary Productivity (NPP) and living plant biomass is low at high latitudes because of harsh climatic conditions and a short growing season, carbon stocks in high-latitude soils, and in particular in permafrost, are very large (e.g. Tarnocai et al., 2009; Hugelius et al., 2011) because of reduced soil organic matter decomposition and the burial of frozen carbon below the active layer over long time scales. Changing soil properties and temperature in response to future warming could therefore release $CO_2$ and $CH_4$ from thawed permafrost, with a potential carbon release on the order of 92 ± 17 PgC by 2100 under the current warming trajectory (RCP8.5) (Schuur et al., 2015). Altogether, high-latitude vegetation significantly affects regional and global climates and must therefore be correctly represented in ESMs, in particular in the light of projected strong Arctic and sub-Arctic climate warming and related biogeographic shifts. With the current warming trajectory, the colonisation of shrubs could be significant (Frost and Epstein, 2014), and as observed by Blok et al. (2011b), it could lead to an Artic greening (Blok et al., 2011b; Bonfils et al., 2012) with increased leaf area, decreased surface albedo in winter, and potential increase of temperatures at local and regional scales.

Until recently the description of circumpolar vegetation in land surface models is relatively simple and discretized on few Plant Functional Types (PFTs) that share similar equations and differ only by parameters values (except for phenology which is usually PFT-specific). In most land surface models (for instance those used in CMIP5 Earth System Models) all vegetation types were classified as either trees or grasses PFTs. Taiga and tundra, where non-vascular plants and shrubs dominate the landscapes in the reality, cover about 15% of global land surfaces (Beringer et al., 2001). In the BIOME ecosystem model (used specifically to study past and future vegetation transition) the tundra diversity was taken into account in the early 2000s (Kaplan et al., 2003). Chadburn et al. (2015) recently included mosses in the JULES model (Best et al., 2011). Similarly, a first description of lichen and bryophytes was implemented in the JSBACH model (Porada et al., 2013). Biogeochemical and biophysical characteristics of shrubs are already implemented in some models, such as in the Community Land Model (Oleson et al., 2013), JULES (Clark et al., 2011) and JSBACH (Baudena et al., 2015). In this study we further develop the ORCHIDEE model (Krinner et al., 2005), the land surface component of the Institute Pierre Simon Laplace (IPSL) ESM, to represent non vascular plants, arctic shrubs and tundra grasses. This study focuses on the parameterizations of these three new PFTs, their interactions part of the Dynamic Global Vegetation Model (DGVM) of ORCHIDEE being treated in a following study.

To date, the ORCHIDEE model contains 8 different types of trees (tropical broad-leaved evergreen and raingreen, temperate needleleaf evergreen, broad-leaved evergreen and summergreen, boreal broad-leaved



summergreen, needleleaf evergreen and summergreen), 4 types of grasses (C3 and C4 grassland as well as C3 and C4 generic crops) and bare soil (Krinner et al., 2005), using the PFT concept. In the reality, high latitude vegetation contains graminoid tundra, shrubs and wetlands including mosses and sedges (see CAVM, for Cirumpolar Arctic Vegetation Map, Mapping Team et al., 2003) while in ORCHIDEE it was represented by a

single PFT for C3 grasses and several PFTs for boreal trees, namely boreal broadleaved deciduous, needleleaved deciduous and evergreen conifers (Krinner et al., 2005). In view of the diversity of circumpolar vegetation, the current discretization of the vegetation in ORCHIDEE does not allow to properly model the regional dynamics of water, carbon and energy fluxes.

Key plant functional types missing in the model for the high-latitudes are mosses and lichens and shrubs.

Mosses and lichens are non-vascular plants; their uptake of nutrients is not supported by xylem sap flow and their gas exchange of water and $CO_2$ is not regulated by stomata. Moreover, mosses and lichens have different environmental needs than grasses (i.e., more resistant for hydric and thermal stress or for nitrogen limitation). Shrubs are smaller than trees and have a different morphology, inducing a larger snow accumulation in winter, and tolerance to wind or cold temperature, and a different potential for colonisation (shrubs being endemic in

many tundra ecosystems can grow rapidly in response to warming whereas trees need to establish).

The aim of this study is to improve the description of circumpolar vegetation in ORCHIDEE by adding mosses and shrubs and adjusting parameters related to C3 grasses, in order to improve the spatial and temporal dynamics of biogeochemical and biophysical processes in the soil-plant-atmosphere continuum. The implementation of the new plant functional types is described in Sect. 2. Results obtained both for site scale

and large-scale simulations are described in Sect. 3. Sect. 4 presents a summary of the key findings together with some perspectives.

## 2       Methods

### 2.1      ORCHIDEE: overall model description

ORCHIDEE describes the exchange of energy, water and carbon between the atmosphere and the biosphere.

The model includes the representation of carbon and water exchange at leaf scale scaled up to canopy-scale, the allocation of carbon within plant compartments (leaves, roots, heartwood and sapwood), autotrophic respiration, litter production, plant mortality and decomposition of soil organic matter (after Parton et al., 1988). Leaf-scale photosynthesis follows the formulation of Farquhar et al. (1980) for C3 plants and Ball and Berry for stomatal conductance (Ball et al., 1987) implemented according to Yin and Struik (2009) and Kattge

and Knorr (2007), i.e. with a seasonal acclimation of maximum photosynthetic rates to temperature.

The soil hydrology model includes an 11-layer diffusion model following the van Genuchten (1980) equations for texture-dependent hydraulic saturation capacity and vertical diffusivity (de Rosnay et al., 2002). The model runs at half-hourly time step but describes slow processes such as carbon allocation, respiration, phenology or litter decomposition at a time step of one day. ORCHIDEE uses the concept of Plant Functional Types (PFTs)

to describe the heterogeneity of land surface ecosystems. Thirteen PFTs (including bare soil) are already present with 8 types of trees and 2 natural and 2 agricultural herbaceous (C3 and C4) types (Krinner et al., 2005), as summarized in Table 1.





The high latitude version of ORCHIDEE (ORC-HL from ORCHIDEE rev1322) used in this study includes a soil-freezing scheme (Gouttevin et al., 2012) and a 3-layer explicit snow model (described initially in Wang et al., 2013). In this new ORCHIDEE version (ORC-HL-VEGv1.0), 3 new PFTs are added to the 13 original ones (Table 1), i.e. non-vascular plants (NVP) including bryophytes (mosses, liverworts and hornworts) and lichens,
boreal shrubs, and boreal C3 grasses. Note that tropical trees are not present in high latitudes.

### 2.2    Non Vascular Plants (NVP): Bryophytes & Lichens

Bryophytes and lichens (NVPs) are very specific plant vegetation types, with a rather small amount of living biomass, around 200 g.m$^{-2}$ (Bond-Lamberty and Gower, 2007; Gornall et al., 2007), but with significant dead organic matter beneath. In contrast, in boreal and tundra ecosystem where mosses compose a small fraction of
total ecosystem biomass, their net primary productivity (NPP) can be up to 50% of total annual NPP (Viereck et al., 1986; Beringer et al., 2001) and corresponds to approximately 1–6% of the global terrestrial net primary productivity (NPP) (Ito, 2011; Porada et al., 2013). In addition, NVPs have no sap (i.e. no water circulation), no roots (only rhizoids to hold on to the ground) and no active stomata to optimize the uptake of $CO_2$ in order to minimize water loss.

We modified the equations of C3 grasses plants in order to describe NVPs as follows. First, we consider that NVP biomass is mainly represented by leaf carbon (i.e., no wood, reserves and root). Their leaves are assumed to access water in the top-soil without roots (i.e. no carbon allocated to a root compartment). We also modified the equations of photosynthesis and stomatal conductance, carbon allocation, and energy balance. In the
following we detail how the few key processes of ORCHIDEE have been adapted as well as the new processes were implemented to represent NVP specificities. For all other processes and associated parameters not described below, we used the C3 grasses equations (as reported by Krinner et al., 2005).

#### 2.2.1    Photosynthesis and stomatal conductance

Photosynthesis of C3 plants in ORCHIDEE is based on Farquhar and Sharkey (1982), with the stomatal conductance ($g_s$) implemented according to Yin and Struik (2009):

$$g_s = g_o + \frac{A+R_d}{C_i + C_{i*}} \times f_{VPD} \qquad (1)$$

With $g_O$ the stomatal conductance when irradiance is null, $A$ the rate of $CO_2$ assimilation, $R_d$ the dark respiration rate, $C_i$ the intercellular $CO_2$ partial pressure and $C_{i*}$ the $C_i$-based $CO_2$ compensation point in the
absence of dark respiration. $f_{VPD}$ is a function describing the effect of leaf-to-air vapour pressure difference ($VPD$), described empirically following Yin and Struik (2009):

$$f_{VPD} = \frac{1}{[1/(a_1 - b_1.VPD) - 1]} \qquad (2)$$

With $a_1$ & $b_1$ empirical constants. This function limits the stomatal conductance under dry air conditions.

Vascular plants have stomata (Kirkham, 2005; Ruszala et al., 2011) to regulate gas fluxes (i.e. $CO_2$, transpiration). For NVPs, the situation is more complex and diverse (Williams and Flanagan, 1996; Chater et al., 2013): some species have "non active" stomata (Ruszala et al., 2011) like *Oedipodium*, others have only



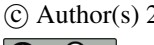

"pseudo-stomata" like *Sphagnum*, and some have no stomata like *Andreaeobryum* (Haig, 2013). For the sake of simplicity and given the lack of a well established photosynthesis model for each NVPs type, we considered that all NVPs have "pseudo-stomata". We thus kept Eq. (1) for $g_s$ (Yin and Struik, 2009) but with a conductance that only weakly depends on the VPD. Observation of NVPs transpiration suggests that their

conductance has a small dependence to humidity and atmospheric $CO_2$ concentration, but a large mean value. We thus defined the coefficients $a_1$ and $a_2$ (see Table 2) so that the VPD dependency of leaf stomatal conductance $f_{vpd}$ in Eq. (2) is almost independent of **VPD** and chose a large value for $g_0$ to simulate a high stomatal conductance. This solution is close to that used by Dimitrov et al. (2011), i.e. a constant conductance.

### 2.2.2    Plant carbon allocation

ORCHIDEE has five biomass carbon reservoirs for C3 grasses: leaves, root, reserve, reproductive organs (fruits), and sapwood below and above ground. We choose to keep only the leaf reservoir to represent the NVP biomass and the fruits pool for reproduction (see Table 2). Furthermore, C3 grasses are summergreen vegetation with only reserve pools during wintertime. Using the leaf pool to represent NVPs biomass implies to consider NVPs as an evergreen PFT (see Table 2) with leaves present all year long. The main challenge is

then to adapt the leaf biomass turnover in order to represent the observed temporal dynamic of lichens and bryophytes biomass.

### 2.2.3    Biomass carbon turnover

We modified the original leaf senescence parameter from 120 days (for grasslands) to 470 days for NVPs (Table 2). Then we defined an energy cost (i.e. an extra turnover of biomass) for NVP survival in cold winter

conditions and limited photosynthesis due to the thickness of the NVPs reducing light penetration. These two processes are described hereafter.

Bryophytes and lichens have a very good resistance to extreme conditions. This adaptation has however an energy and thus a biomass cost, modelled through an additional carbon loss ($t_{npp0}$ in gC.m$^{-2}$.d$^{-1}$) based on the

cumulative number of day ($d_{cum}$) when the Net Primary productivity (NPP) is negative or null, as given by Eq. (3).

$$t_{npp0} = b \times k_l$$

$$k_l = \begin{cases} 0 & , \ d_{cum} < d_0 \\ k_{l\,max} \times \frac{(d_{cum} - d_0)}{(d_m - d_0)}, & d_0 < d_{cum} < d_m \\ k_{l\,max} \times \frac{(d_{cum} - d_f)}{(d_m - d_f)}, & d_m < d_{cum} < d_f \end{cases} \tag{3}$$

Where **b** the (leaf) biomass of NVPs (gC.m$^{-2}$) and $k_l$ the additional fraction of biomass lost during extreme conditions (or turnover rate in d$^{-1}$) with a maximum value of $k_{l\,max}$ (in d$^{-1}$), $d_0$ a threshold delay time (in days)

before increasing the turnover, $d_f$ (days) the maximum number of days for applying the extra turnover, and $d_m$ (days) the day number when $k_l$ reaches its maximum value after $d_0$. The values of all parameters are summarised in Table 2. Figure 1 illustrates the increasing biomass turnover linked to extreme conditions with $k_l$ as a function of time in the season with negative or zero NPP.

Using NPP to determine the period of the year with extreme conditions allows us to combine different stress

factors such as cold temperature and very low moisture. Hence the combination of short-term stress episodes





(periods when $d_0 > 0$) such as a short drought followed by a snowfall (blocking of light and cold temperatures stress) on the NVPs could result in a long-term impact (increase in turnover) on vegetation.

The second turnover is related to favourable conditions with a large growth of biomass during the growing season (such as in peatlands). Given their large NPP under favourable conditions, NVPs can accumulate biomass over several tens of centimetres. In this case, sunlight cannot reach the lower portion of the canopy due to light penetration decreasing, although this biomass is still considered as leaf material (see 2.2.1). The underneath biomass usually dies from a lack of light and possibly a lack of oxygen in wet conditions. Given that oxygen concentration is not simulated in this model, the effect of anoxic conditions and severe light limitation are simply parameterized by increasing the overall leaf biomass turnover rate during the growing season. We chose the Leaf Area Index (LAI) to define this additional turnover: when the maximum LAI ($LAI_{max}$) is reached, the underlying layers will not receive any sunlight, resulting in an increase of their turnover ($t_{missL}$) represented by Eq. (4).

$$t_{missL} = b \times \left( e^{l_{coef} \times (LAI - LAI_{lim})} - 1 \right), \text{ if } LAI > LAI_{max}, \tag{3}$$

Where $b$ is the daily leaf biomass of NVPs (gC.m$^{-2}$), $l_{coef}$ a coefficient and $LAI_{lim}$ a threshold leaf area index defined from Bond-Lamberty and Gower (2007). These two parameters are optimized in Sect. 2.6.1 and their values reported in Table 2.

### 2.2.4 Water access for NVPs

**Plant water uptake**

In ORCHIDEE, all vegetation types have access to soil water through a root system. The ability of roots to extract water depends on soil moisture in the different soil layers (11 currently, see 2.1) and the root density profile (R) (de Rosnay, 1999):

$$R(z) = e^{-r_p \times z} \tag{5}$$

With $z$ the soil depth (m) and $r_p$ a PFT dependent parameter to control the shape of the root profile.

NVPs do not have roots to absorb water (or nutrients from the underlying substrate). Some of them, such as *Sphagnum*, can have threadlike rhizoids but only to anchor to the soil. So they can only access the surrounding surface water. However, ORCHIDEE does not include a surface liquid water reservoir; thus for simplicity we have assumed that NVPs have access to water stored in the first top-soil layers. This assumption allows keeping an internal coherence between PFTs and facilitating the treatment of the competition for water between PFTs. The value of the $r_p$ parameter (Table 2) for NVPs was defined through the optimization (see Sect. 2.6.1). With 50% water uptake (without roots) at 2.5cm and 95% at 11cm, we obtained water access values closed to those proposed by Dimitrov et al. (2011) or by Chadburn et al. (2015). Figure 2 illustrates the soil water uptake profile for NVPs, and the root profiles for C3 grasses and boreal trees (use in ORCHIDEE).

**Impact of drought on the desiccation of NVPs**

During and after a water stress period, the water content of NVPs decreases significantly (desiccation) which reduces the plant photosynthetic capacity (Williams and Flanagan, 1996; Wania et al., 2009; Dimitrov et al., 2011). As for the other PFTs in ORCHIDEE, the instantaneous effect of soil water limitation will reduce





photosynthesis through a soil water stress function imposed on the maximum photosynthetic capacity (Farquhar et al, (1980) photosynthesis model). Additionally, for NVPs, plant desiccation occurs and the time needed before recovery to optimum photosynthetic capacity must be taken into account.

To account for this effect, Wania et al. (2009) reduced gross primary production as a function of the annual mean water table position. In ORC-HL-VEGv1.0 we chose to use a monthly running mean hydric stress factor ($w_s$) computed from the relative water content in each soil layer weighted by the specific water uptake profile of NVPs defined in Fig. 2. We defined a desiccation function, $d_{ess}$, as a linear function of $w_s$ (Eq. (6) and Fig. 3) varying between 1 (no impact) and a minimum value $d_{off}$, when $w_s$ decreases to zero under maximum water

stress. The function $d_{ess}(w_s)$ illustrated in Fig. 3 scales the maximum rate of carboxylation ($Vc_{max}$) as well as the maintenance respiration. The maximum rate of electron transport ($Vj_{max}$) is scaled through $Vc_{max}$. Indeed, leaves maintenance respiration defined in ORCHIDEE being a function of the leaf carbon content (biomass) and LAI, should then be reduced when NVP get desiccated. With this formulation, we can take into account the impact of a drought on a monthly time scale.

$$d_{ess} = \begin{cases} d_{off} + \frac{1-d_{off}}{w_{s\,min}} \times w_s, w_s < w_{s\,min} \\ 1 \qquad\qquad , w_s \geq w_{s\,min} \end{cases} \qquad\qquad (6)$$

With $w_{s\,min}$ the minimum threshold hydric stress for desiccation (a constant defined in Table 2).

### 2.2.5    Heat transfers

Non vascular plants and more precisely bryophytes form an insulating layer above the soil, with thus a strong control on the heat exchange between the atmosphere and the soil (Dyrness, 1982; Beringer et al., 2001; Blok

et al., 2011a). In its standard version, ORCHIDEE does not account for the thermal insulation properties of vegetation in the calculation of the surface energy budget. For the sake of simplicity and following the same approach as in Chadburn et al. (2015), we modified in ORC-HL-VEG the upper soil layer characteristics to describe the effects of NVPs on the heat transfers to the soil over a depth that is equivalent to the NVP thickness and for the fraction of each grid cell box covered by NVPs.

First we estimate the thickness of NVPs ($h$) assuming a fixed biomass density:

$$h = \frac{b}{\rho} \qquad\qquad (7)$$

With $b$ the total NVPs biomass (g.m$^{-2}$) and $\rho$ its density (gC.m$^{-3}$; see Table 2)

The thermal capacity / conductivity (Eqs. (8) & (9)) of the upper soil layers (equivalent to the depth of the

NVPs layer) are modified based on the soil volumetric moisture content (as in the standard ORCHIDEE version) and the heat conductivity and capacity of NVPs, following Soudzilovskaia et al. (2013). The heat thermal capacity of the top-soil thickness $h$ occupied by NVPs, $C$, follows from:

$$C = C_{dry} + m_{vol} \times (C_{wet} - C_{dry}) \qquad\qquad (8)$$

Where $m_{vol}$ is the volumetric relative moisture content over a thickness $h$, $C_{dry}$ the dry thermal capacity of dry

NVPs and $C_{wet}$ the wet heat capacity of wet NVPs (from Soudzilovskaia et al., 2013; see Table 2). Note that in the standard case without NVPs, $C_{wet}$ and $C_{dry}$ are defined from the soil texture (see Wang et al., 2016). In the case of frozen soil we use an ice capacity ($C_{ice}$) for NVPs, deduced relatively to $C_{ice}$ of soil. When the soil is





partly frozen a weighting average between the two thermal capacities is calculated (using, $x$, the unfrozen soil fraction). The overall thermal conductivity, $\lambda$, follows from:

$$\lambda = \lambda_{dry} + m_{vol} \times (\lambda_{sat} - \lambda_{dry})$$
$$\lambda_{sat} = \lambda_{sat\_wet}{}^{x} \times \lambda_{sat\_ice}{}^{1-x} \tag{9}$$

With $\lambda_{dry}$ the dry soil thermal conductivity, $\lambda_{sat\_wet}$ the unfrozen wet thermal conductivity (from Soudzilovskaia

et al., 2013) and $\lambda_{sat\_ice}$ the frozen thermal conductivity of NVPs (derived relatively to $\lambda_{sat\_ice}$ of soil.) See Table 2 for values and units. Note that the current version of ORCHIDEE only calculates one energy budget being the average of all vegetation types present in a grid cell; the overall thermal soil characteristics thus correspond to a weighted average of the soil characteristics according to the fraction of NVPs covering a grid cell.

### 2.2.6    Soil organic matter decomposition

In the standard version of ORCHIDEE, two important factors, temperature and moisture, exert a control on litter and soil organic matter decomposition (following the CENTURY model, Parton et al., 1988). These factors are computed from weighted mean soil temperature and soil moisture profiles, assuming an exponential profile of soil organic matter content and associated decomposition processes between 0 and 2m depth. For the moisture control of decomposition, the original function (Parton et al., 1988; Krinner et al., 2005) is increasing

with soil moisture content (maximum at saturation), which is not adapted for water-saturated soils, where anoxic condition reduces bacterial activity (such as in peatlands). As these conditions may prevail for NVPs covers, we modified the original scheme.

First, we introduced a vertical discretization of below ground litter carbon pools, assuming it follows the same distribution as the root profile for vascular plants or soil water uptake profile for NVPs (exponential decay as

Eq. (5), in de Rosnay, 1999), as in Frolking et al. (2001). Thus, the below ground litter is considered to be linked to vegetation source (i.e. roots for vascular plants). Moreover we consider that there is no above-ground litter for NVP, so that leaf litter is treated like below ground litter, as in Frolking et al. (2001) and Chadburn et al. (2015). With this new vertical discretization, we chose to use the temperature and soil moisture of each layer to define the control litter decomposition.

To account for anoxic conditions often prevailing in water saturated NVP ecosystems causing slow decomposition rates (Frolking et al., 2001), we changed the moisture decomposition function ($R_{SR}$) applied for each layer as in Moyano et al. (2012), using a look-up table approach. Equation (10) describes the new function and the reduced decomposition with soil moisture content (applied for the litter issue of all PFTs).

$$PR_{SL}(m_{vol}) = m_{c(3)} \times m_{vol}{}^{3} + m_{c(2)} \times m_{vol}{}^{2} + m_{c(1)} \times m_{vol} + m_{c(0)}$$

$$SR\,(m_{vol}) = \prod_{k=0}^{m_{vol}} PR_{SL}(k)$$

$$R_{SR}(m_{vol}) = \frac{SR(m_{vol})}{\max_{0<k<1}(SR(k))} \tag{10}$$

With $SR$ the soil respiration (coefficient), $PR_{SR}$ the proportional response of $SR$ to soil moisture, $R_{SR}$ the relative respiration, $m_{vol}$ the soil volumetric moisture content (unit less), $m_{c(1-3)}$ three parameters taken from Moyano et al. (2012). $SR$ is equal to the product of all $PR_{SL}$ values (denoted by $\Pi$ symbol) at each 0.01 moisture interval ($k$), from zero to the computed $SR$ moisture. To obtain $R_{SR}$, $SR$ is divided by the maximum of



*SR* for all *k* intervals (0 to 1). See Table 2 for constant values. Note that the temperature function decomposition is not modified.

### 2.2.7 Summary and other parameters

Other parameters and processes used for NVPs are set equal to those of C3 grasses, such as albedo and roughness as described by Krinner et al. (2005). We have optimized specific parameters of NVPs (listed with asterisk in Table 2) against observation (see Sect. 2.5.1), following a Bayesian optimization framework (see Sect. 2.6.1). The values of the main parameters for the NVPs including the optimized ones are reported in Table 2.

The implementation of the NVP PFT is performed in such a way that if we need to separate in different sub-PFTs (i.e. study bryophytes and lichens separately), this would be easy to do, with new associated parameters.

### 2.3 Boreal deciduous shrubs

Shrubs share similar biogeochemical and biophysical processes as trees. Therefore, the introduction of a new shrub PFT is based on the equations of the boreal deciduous broadleaved tree PFT. The main difference between trees and shrubs concerns the size, and thus the allometry resulting from carbon allocation. Further, shrubs grow faster and therefore colonize landscapes before trees do. For high-latitudes, the protection of shrubs against cold by snow is an important process that needs to be taken into account, since snow depth and shrub height are positively correlated (McFadden et al., 2001; Sturm et al., 2001). Snow cover tends to be thicker when shrubs are present (McFadden et al., 2001), and a thicker snow cover better protects shrubs from frost damage.

In the following, we describe these particularities including the new allometry, the snow – shrubs interactions as well as the impact of shrubs on surface roughness and albedo. Note that all modifications are generic so that we can easily create additional shrubs types, such as needleleaf or evergreen phenotype, with only few parameter changes.

### 2.3.1 Shrub allometry

Tree allometry in ORCHIDEE is based on a pipe tune model (Smith et al., 2001). It represents the relation between height and diameter as a power (or log-linear) function, with no height limit. Shrub development is more horizontal than vertical (Bentley et al., 1970; Sitch et al., 2003; Lufafa et al., 2009), which requires modification of the tree allometry. We implemented the allometry rules described by Aiba and Kohyama (1996) with specific values for shrubs from Martínez and López-Portillo (2003). Equation (11) gives the allometry relation between individual height ($H$, m), diameter ($D$, m), volume ($V$, m$^3$), the number of individuals ($n_i$), the total crown area ($C_a$, m$^2$), the total stem basal areal ($T$, m$^2$), the total woody biomass ($m_w$, gC.m$^{-2}$) and wood density ($\rho_w$, between 0 and 1). The height of a shrub is a logarithmic function of its diameter (Eq. (11.a)) and its volume is represented by a cylinder (Eq. (11.b)). The shrub vegetation cover is defined as a function of the total stem basal area (Eq. (11.c)). With simple geometric relations (Eq. (11.d)) and assuming a fixed crown area ($C_a$ become a constant) the system can be solved and all key variables expressed as a function





of shrub woody biomass ($m_w$) (the height is given by Eq. (11.e)). If the crown area is not fixed (e.g. with dynamical vegetation), there is no analytical solution to obtain the height.

a) $1/H = 1/(A \times D^\gamma) + 1/H_{max}$

b) $V = \pi/4 \times \frac{H_{max} \times A \times D^{2+\gamma}}{H_{max} + A \times D^2}$

c) $C_a = \beta \times T^\alpha = \beta \times \left(n_i \times \pi/4 \times D^2\right)^\alpha$

d) $m_w = n_i \times V \times \rho_w$ and $H = \frac{n_i \times V}{T}$

e) $H = \frac{m_w}{\rho_w \times \left(C_a/\beta\right)^{1/\alpha}}$ (11)

Were $A$, $\beta$, $\gamma$, $\alpha$ and $H_{max}$ are parameters adapted from Martínez and López-Portillo (2003) (see Table 4). Here, the parameter $H_{max}$ defining the maximal height (m) was optimized (see Sect. 2.6.1). To be in accordance with imposed vegetation coverage, a minimum woody vegetation height ($H_{min}$, m) was prescribed, based on the maximum height, according to:

$$H_{min} = H_{max}/h_c$$ (12)

Where $h_c$ is a factor defined in Table 4. Based on the new shrub allometry description equations (Eq. (11)), new parameters can be derived for shrubs with the pipe tune model (Table 4).

### 2.3.2 Impact of shrubs on snow

Shrub vegetation affects snow cover through snow compaction and spatial heterogeneity of snow deposition. Shrub (and tree) branches support part of the snow cover. As a result, the snow weight on lower snow layers is smaller and the compaction of snow crystals is reduced. Moreover, wind is reduced by the presence of a shrub (and tree) canopy, which further reduces snow compaction compared to short vegetation cover. We kept the original snow compaction equation in ORCHIDEE (Wang et al. (2013), their Eqs. (11), (12) and table A1) but chose new values for the parameters controlling compaction depending upon low or high vegetation (Table 3) in order to model a different depth and density over the fraction of a grid cell covered with shrubs (and tree). Currently there is no sub-grid simulation of snow cover and energy balance in ORCHIDEE, so there is no distinction according to the fraction of different PFTs present in a grid cell. To account for differences between PFTs we compute snow compaction separately for short vegetation (bare soil, grass and NVP), shrubs and trees. The resulting average snow depth and density over a grid cell is obtained by weighting each vegetation-dependent compaction by its fraction. The deposition of snow is assumed to be the same among the different PFTs. A PFT dependent snow depth is needed to compute the protection of vegetation by snow (Sect. 2.3.3). To compensate for the lack of an explicit PFT dependent snow depth, an empirical correction is applied to account for the effect of vegetation type on snow compaction and deposition on shrubs:

$$d_{s\_v} = d_{s\_f} \times d_s$$
$$d_{s\_f} = \begin{cases} 1 + f_v, & f_v \leq 0.5 \\ 2 - f_v, & f_v > 0.5 \end{cases}$$ (13)

With $d_{s\_v}$ the snow depth of high vegetation (shrubs and trees, m), $d_s$ the average snow depth (m) over the grid-cell, and $d_{s\_f}$ a function of $f_v$, the fraction of high vegetation.





### 2.3.3 Shrubs mortality reduced by snow protection

ORCHIDEE, when used to compute dynamically the vegetation distribution includes a tree mortality during extremely cold days, calculated as the percentage of biomass lost at the end of each day (see Zhu et al., 2015). This mortality depends on a minimum temperature, as defined in Eq. (14). We used the same equation than in

Zhu et al. (2015) but boreal needleleaf trees are also assigned a critical minimum survival temperature.

If $T_{min} < T_{min,crit}$, $M_{ce} = k_{ce} \times (T_{min,crit} - T_{min})$ (14)

With $M_{ce}$ the mortality rate due to cold extremes, $T_{min,crit}$ the minimum critical survival temperature (defined for each PFT), $T_{min}$ the daily minimum air temperature and $k_{ce}$ a mortality coefficient. The values of these parameters are given in Table 4.

For shrubs we use a similar approach to control the loss of biomass due to extreme cold temperature. A mortality rate similar to Eq. (14) is applied to the highest parts of shrubs that are not covered by snow. For the part of shrubs situated inside snow layers (see 2.3.2, Eq. (13) for the shrubs snow depth calculation), snowpack temperature is used in Eq. (15). We defined a daily vertical profile of minimum temperature $T_{min}(z)$ function of shrub height above ground ($z$), by linear interpolation between soil, snow layers and air temperatures above the

shrub height emerging from the snow pack. To simulate the mortality of shrub parts being exposed to extreme cold, the following mortality equation is applied from the top part of shrubs.

$$M_{ce} = \int_{H_{min}}^{H} k_{ce} \cdot fn(T) \, dz$$
$$f_n(z) = \begin{cases} 0 & , T_{min} \geq T_{min\,crit} \\ T_{min\,crit} - T_{min}(z), & T_{min} < T_{min\,crit} \end{cases}$$ (15)

With $M_{ce}$ the extreme cold mortality, $T_{min.crit}$ a minimum critical temperature (defined by PFT), $k_{ce}$ a

coefficient, $H$ is the shrub height and $H_{min}$ its minimum height (Eq. (12)). The values of the parameters of Eq. (15) for shrubs are given in Table 4. This equation is the integral of Eq. (14) applied to the height of shrubs.

### 2.3.4 Modification of roughness and albedo

In ORCHIDEE the surface roughness length is directly computed from the height of the vegetation. Similarly, surface albedo depends on the vegetation type. Because shrubs can be partially or entirely covered by snow,

the computation of surface roughness and albedo in the presence of shrubs needs to take into account snow height. The calculation of surface roughness length has thus been modified. First vegetation height is computed separately for shrubs (using Eq. (11)) and for trees (using the original pipe tune model equation of Smith et al., 2001). The height of the snow cover over shrubs is then subtracted from the vegetation height in order to estimate the height of the vegetation above the snow surface (i.e. the relative height), which determines the

surface roughness. The relative difference between the relative height and the total height is not substantial for trees (height > 5m), but it can be important for shrubs (> 30cm) which can be totally covered by snow. To represent the spatial heterogeneity of snow cover, when the snow thickness is close to the height of vegetation, a linear function is applied to estimate the height above snow:

$$H_{PFT_{as}} = \begin{cases} H_{PFT} - d_s & , H_{PFT} > d_s \cdot (1 + \Delta_{z_0}) \\ 0 & , H_{PFT} < d_s \cdot (1 - \Delta_{z_0}) \\ (H_{PFT} - d_s (1 - \Delta_{z0}))/2, & \text{else} \end{cases}$$ (16)

Where $H_{PFT}$ is the height of the PFT, $H_{PFT\_as}$ is the height of the PFT above the snow, $d_s$ depth of snow, and $\Delta_{zo}$ the width of the transition zone due to spatial heterogeneity of snow cover (see Table 4).



To compute the roughness length $z_0$, for trees and shrubs the maximum fraction of vegetation $f_v = f_{v\_max}$ (prescribed if the vegetation cover is static, or calculated when the vegetation cover is dynamic, and independent of LAI) is used to take into account the influence of trunks and branches even if there are no

leaves. For grasses and NVPs, only the projected surface of the foliage in the canopy $f_v = f_{v\_max}(1-e^{-LAI/2})$ is used because there is no woody elements. The rest of surface is considered as bare soil with a constant roughness length value.

Finally the roughness length of a given PFT is calculated as its height above snow multiplied by a roughness parameter $z_{o\_c}$, as initially in ORCHIDEE. If this value is lower than the bare soil roughness ($z_{0\_bs}$ fixed), then

the latter value is used. The grid cell mean roughness length is computed as a function of each PFT roughness weighted by the vegetation cover, $f_v$:

$$\log(z_0) = \sum_{PFT} \left( f_v \times \log \left( \max \left( \frac{H_{PFT\_as}}{z_{o\_c}}, z_{o\_bs} \right) \right) \right) \tag{17}$$

Where $z_0$ is the grid-cell averaged roughness (m), $z_{0\_bs}$ the roughness of the bare soil (m), $f_v$ the fraction of each PFT and $z_{0\_c}$ a constant roughness parameter. The values of the parameters of Eq. (17) are given in Table 4.

The mean albedo of a grid cell depends on the vegetation, bare soil and snow albedo and their fractional coverage. While snow albedo is a function of snow age (computed for each vegetation type), bare soil and vegetation albedo are constant in time. A critical parameter to weigh the different terms is the fraction of the grid cell covered by snow, $snow_{frac}$, on bare soil and vegetation. In ORCHIDEE this fraction only depends on

the snow mass, as defined in Chalita and Le Treut (1994). We chose to modify this approach in order to account for the effect of the vegetation structure as in Douville et al. (1995) and Boone (2002), using the roughness length calculated from Eq. (17) which is given by:

$$snow_{frac} = \frac{snow_{dz}}{snow_{dz} + \xi.z_0} \tag{18}$$

With $snow_{frac}$ the fraction of the grid covered by snow, $snow_{dz}$ the snow thickness, $z_0$ the roughness length and

$\xi$ a parameter (defined in Table 4).

### 2.3.5    Shrub parameters

Table 4 summarizes the main parameter values used in the equation described previously as well as few other parameters modified for the shrub PFT (compared to the initial tree PFT).

### 2.4    Cold climates C3-Grasses

We re-parameterized the grassland PFT for circumpolar regions, following the generic equations of C3 grasses. Few parameters have been calibrated (see list in Table 5) to modify primarily the photosynthetic activity, the root distribution in the soil and the leaf development.

The rate of carboxylation limited by Rubisco ($V_c$) and by electron transport ($V_j$) are dependent on specific parameters (following Yin and Struik (2009) and presented in Eq. (19)), themselves function of monthly mean

temperature ($t_m$, K) (Eq. (20)).



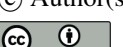

$$F(T) = k_{25} \times e^{\frac{E_a(T-T_{25})}{T_{25}.R.T}} \times \frac{1+e^{\frac{T_{25}.\Delta S - E_d}{T_{25}.R}}}{1+e^{\frac{T.\Delta S - E_d}{T.R}}}$$  (19)

With F(T) the rate function $V_c$ or $V_j$, $k_{25}$ the maximum of each rate ($Vc_{\_max}$ or $Vj_{\_max}$) at a reference temperature $T_{25}$ (25°C or 298 K; note that $Vc_{\_max}$ and $Vj_{\_max}$ are linked by a linear function being temperature dependent), $T$ the current temperature (K), $E_a$ the activation energy, $E_d$ the deactivation energy, $\Delta S$ the entropy factor and $R$ the ideal gas constant (Table 5).

The entropy factor $\Delta S$ for $Vc_{max}$ or $Vj_{max}$ is calculated as follows:

$$\Delta S = a + b \times t_m$$  (20)

With $a$ and $b$ two constants (Table 5). This formulation from Kattge and Knorr (2007) include an adaptation of seasonal growth temperature (derived from spatial relation between $Vc_{max}$ and $J_{max}$ in TRY database and extrapolated for temporal equations). Observations by Miller and Smith (2012) of the optimal temperature for photosynthesis for graminoids and forb tundra (10 to 20°C) were used to define new parameter values, which were then optimized (list of variable in Table 5). The optimization procedure is described in Sect. 2.6.1.

The depth over which 95% of the root is located corresponds roughly to 0.5 meter for boreal C3 grasses and to 1 meter for temperate C3 grasses, according to Bonan et al. (2003) or Iversen et al. (2015). Using this estimate we changed the a priori value of the root profile shape parameter ($r_p$ parameter; see Eq. (5) and de Rosnay, 1999) for cold grasses and after optimization (see Table 5) we obtained that 95% of the roots are within the first 40 cm of the soil.

The specific leaf area (SLA) was also optimized for cold climate grasses, using as a priori the initial values from C3 temperate grasses. Note that for simplicity and because of their weak impact on simulation when the vegetation is fixed, we did not add survival or establishment limits as in Bonan et al., (2003) and Oleson et al. (2013) or a cumulated degree-day threshold (above zero criteria) for the development (Miller and Smith, 2012).

## 2.5    Observations and vegetation distribution

### 2.5.1    Field survey data

The calibration of the parameters entering in the equations of NVP, shrubs and cold climate grasses is based on observations for the period 1993-2001 gathered in Peregon et al. (2008) and extended up to 2013 for this study. The data set contains georeferenced point-scale observations of the total summertime living biomass (g.m$^{-2}$) and annual net primary productivity NPP (g.m$^{-2}$.yr$^{-1}$) for non-vascular plants (mosses and lichens) and vascular plants (grasses and shrubs) in boreal wetlands. Test sites for field observations located in Western Siberia (Latitude 55° to 71° N, Longitude 63 to 91° E), which is suited for spatial analysis of NPP and biomass due to its flat topography along a wide latitudinal gradient and large variety of natural ecosystems, with minor anthropogenic influence.

At each test site, detailed geobotanical descriptions were recorded and biomass sampling was conducted. Sampling was repeated two or three times during the growing season at the same test sites for several consecutive years to obtain information on interannual variability. Field studies were conducted between June and October at more than 99% of the test sites, and between July and September for 90% of them. General





descriptions of in-field and laboratory methods used to estimate NPP and biomass in wetlands are described in Peregon et al. (2008, 2016).

The data set takes into account all components of NPP and living biomass: above-, land-surface and belowground fractions measured in-situ at different topographical features (such as hummocks, hollows, ridges). In order to avoid the "bound" effect and use of values at the border between two vegetation classes, we chose to take into account only observation where the studied vegetation represented at least 10% of the surface. Spatial differences in these microsite characteristics (i.e. hydrologic and thermal regimes, nutrient availability) strongly determine vegetation characteristics, as well as NPP and biomass, and small-scale heterogeneity induced by these microsite characteristics can be as large as the large-scale variability due to climatic gradients across the area covered by the dataset. Because the small-scale variability cannot be represented in a large-scale model like ORCHIDEE, and small-scale information on microsite hydrological and topographical characteristics were not available, no perfect model-data fit can be expected and we should rather seek for a broad model-data agreement.

The data have therefore been grouped into supersites at 0.5° spatial resolution, giving 36 supersites. The 36 sites have data on mosses (comprising in total 1209 individual observations), but only 16 supersites presenting non-vascular plants, shrubs and grasses (comprising in total 660 individual observations) (Fig. 4).

### 2.5.2 Vegetation distribution

For this study we prescribe the spatial distribution of the vegetation, while a follow-up study will focus on the dynamics of vegetation. We thus had to update the vegetation map used by the standard version of ORCHIDEE in order to include the spatial distribution of the new PFTs. The land cover product used to define PFT distribution in ORCHIDEE is derived from the land cover product of the European Space Agency (ESA) Climate Change Initiative (CCI) (available at http://www.esa-landcover-cci.org/). The product is based on medium–resolution satellite observation and provides information on the vegetation distribution using land cover classes (LCC) defined by the United Nations Land Cover Classification System (UNLCCS). In order to match the satellite land cover classes with the PFTs coverage in ORCHIDEE, we use a conversion table established by Poulter et al. (2015). Note that the climate classification system of Köppen (Peel et al., 2007) is also used to further partition some vegetation types into tropical, temperate and boreal zones (see also Poulter et al., 2015). The new vegetation map is thus obtained from this Land Cover dataset (version 1.6.1) transformed by conversion table from Poulter et al. (2015) (tool available from http://maps.elie.ucl.ac.be/CCI/viewer/), from 300m LCC data. From the standard conversion table used in ORCHIDEE, the three new PFTs were included using the following modifications:

  i. The C3 grasses (initially defined globally) that were located in class 5 of Köppen classification (polar and alpine climates) were assigned to the new cold climate C3 grasses PFT.

  ii. In the original version of the conversion table, LCCs were first separated between trees and shrubs (Table S1), then aggregated into trees PFTs. Here we kept the shrubs and trees separated to define the shrub PFT coverage.

  iii. "Lichens and mosses" LCC were classified by Poulter et al. (2015) into C3 grasses and bare soil PFTs, and now are used to define a separate NVP PFT (Table S1). However the NVP coverage that corresponds to the lichens and mosses LCC is clearly underestimated with the CCI product



over Eurasia compared to North America and to other pan-arctic land cover maps (i.e. in Cirumpolar Arctic Vegetation Map: CAVM Mapping Team et al., 2003), in which NVPs cover is much larger. In Loveland et al. (2000) map, we noticed that the tundra biome corresponds to the "sparse vegetation" or to the "lichens and mosses" LCCs distribution; in CAVM Mapping Team et al. (2003) the tundra biome is described as a composite of ~30 to 60% of NVPs. Combining these two maps with the ESA CCI LCC map, we modified the conversion of "sparse vegetation" LCC in the ESA CCI map, initially to 35% bare soil and 40% grass PFTs, into 20% of bare soil, 10% cold climate grass PFT and 45% of the NVP PFT (Table S1).

The resulting spatial distribution of is consistent with CAVM and Loveland et al, with 2.9, 2.2 and 2.8 millions km$^2$ of NVPs, shrubs and cold climate grasses, respectively, north of 60°N.

The distribution of the different circumboreal PFTs is presented in Fig. 5. NVPs are mainly present in northern latitudes where climate conditions for the other PFTs are too extreme. Shrubs are present everywhere in northern latitudes but sparsely, with the tree PFTs always dominating. This is due to the approach we chose, because shrubs are diagnosed from the same LCCs as trees, with a smaller fractional coverage (Table S1). The cold climate C3 grasses come mainly from boreal forest LCCs in northern latitudes and from meadows further south (Table S1). They are dominant only in the latter.

### 2.6 Optimization strategy and evaluation protocol

### 2.6.1 Parameter optimization strategy

We used a Bayesian optimization procedure to optimize selected parameters of the new NVPs, shrubs and boreal C3 grass PFTs, where prior information on the parameter is combined with the information that can be extracted from an ensemble of observations (see Sect. 2.5.1). Assuming that the errors associated with the parameters, the observations and the model follow Gaussian distributions, the optimal parameter set corresponds to the minimum of a cost function, $J(x)$, that measures the mismatch between i) the observations ($y$) and the corresponding model outputs, $H(x)$, (where $H$ is the model operator), and ii) the a priori ($x_b$) and optimized parameters ($x$), weighted by their error covariance matrices (Tarantola, 1987; Eq. (21)):

$$J(x) = \frac{1}{2}\left[(H(x) - y)^T \mathbf{R}^{-1}(H(x) - y) + (x - x_b)^T \mathbf{B}^{-1}(x - x_b)\right] \tag{21}$$

$\mathbf{R}$ represents the error variance/covariance matrix associated with the observations and $\mathbf{B}$ the parameter prior error variance/covariance matrix. Note that $\mathbf{R}$ includes the errors on the measurements, model structure and the meteorological forcing. Model errors are rather difficult to assess and may be much larger than the measurement error itself. Therefore we chose to focus on the structural error and defined the variances in $\mathbf{R}$ as the mean squared difference between the prior model and the observations (as in (Kuppel et al., 2013). For simplicity we assumed that the observation error covariances were independent between the different observations and therefore we kept $\mathbf{R}$ diagonal (off-diagonal terms set to zero).

The determination of the optimal parameter vector that minimizes $J(x)$ is performed using a Monte Carlo approach based on a Genetic Algorithm (GA) following the implementation of Santaren et al. (2014). The algorithm works iteratively, starting with a pool of vectors of parameters (i.e. the chromosomes) defined from randomly perturbed parameters. At each iteration, it randomly perturbs or exchanges parameters of the chromosomes and ranks them based on the cost function values, so that the best chromosomes (parameter



combinations corresponding to the lower cost function values) produce more descendants (following the principle of natural selection). For details of the implementation see Santaren et al. (2014). Note that this algorithm is more efficient to find the minimum of J than a gradient-based method as discussed in Bastrikov et al. (in preparation).

For each optimized parameter, the initial values were taken from the literature or from the values used for the ORCHIDEE boreal deciduous tree PFT for shrubs and from the C3 grasses PFT for NVPs and cold climate C3 grasses. We defined the observation errors (R diagonal) as 50 $gC.m^{-2}$ (1-sigma standard deviation) for the biomass and for NPP, based on field measurements errors (Peregon et al., 2008) and a priori model data mismatch. The number of iteration was set to 25 and the number of chromosomes to 15 for NVPs and 10 for

C3 grasses and shrubs, after some initial check of the convergence of the algorithm. Note that in order to spin up the model with respect to the living biomass, each simulation starts 10 years before the observation period for NVPs and grasses, and 19 years for shrubs.

### 2.6.2    Evaluation Protocol

To illustrate the impact of new boreal vegetation compared to standard PFTs we show the results of two

different simulations: one with the standard 13 PFTs of ORCHIDEE (ORC13) and the second with the new circumboreal PFTs (13 standards + 3 new PFTs: ORC16). Both simulations use the CRU-NCEP meteorological forcing (Wei et al., 2014; Viovy, 2015) based on gridded monthly observations from the Climatic Research Unit (CRU) at 0.5° and the climate re-analysis from the National Center for Environmental Prediction (NCEP) model (reduced to 2° resolution), available from 1901 to 2013. We spin up the model

carbon pools (above and below ground) with a 5,000 years simulation recycling the forcing files from 1901 to 1950 randomly). We then used a transient simulation from 1901 to 2004 with linked $CO_2$ concentration. The spatial domain is also limited to the latitudes above 40° North.

First, the total biomass and NPP are evaluated against observations with extended data from Peregon et al. (2008). We then analyse other key variables (such as LAI, albedo, soil temperature, total evaporation, etc.) to

provide further insight on the impacts on carbon, energy and water fluxes. The analysis is carried out on multiple spatial and temporal scales. Then, to evaluate the simulated LAI, we use the GLASS (Global Land Surface Satellite) LAI product (Liang et al., 2013; Xiao et al., 2014). This product has a temporal resolution of 8 days and is available from 1982 to 2012. Data used in this study cover the period from 2004 to 2013 and were derived from MODIS (moderate resolution imaging spectroradiometer) land surface reflectance

(MOD09A1), at a resolution of 1 km. In order to compare this GLASS product with our 2° resolution simulations, an extrapolated map of the 1 km resolution to the 2° resolution was built and a mask was applied to remove 2° resolution grid cells with a land fraction below 0.7.

## 3      Results

### 3.1     Model calibration and fit to the observations

Following the optimization protocol described in Sect. 2.6.1, we calibrated 12, 6 and 7 parameters for the NVPs, shrubs and cold climate grasses respectively (see list in Table 2, Table 4 and Table 5). The optimization relies on observations of living biomass and Net Primary Productivity observations presented in Sect. 2.5.1.



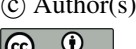

First we should notice that the selected observations are characterized by a very large standard deviation (SD). For cold climate grasses the SDs of the observed total biomass and NPP are close to their mean values (total biomass = 558 ± 427 $gC.m^{-2}.y^{-1}$; NPP = 321 ± 222 $gC.m^{-2}$). For boreal shrubs the SDs are also very large (total biomass = 768 ± 432 $gC.m^{-2}.y^{-1}$; NPP = 321 ± 104 $gC.m^{-2}$), while for non-vascular plants they reach only half

of the mean values (total biomass = 217 ± 105 $gC.m^{-2}.y^{-1}$; NPP = 117 ± 61 $gC.m^{-2}$). The cost function (J($\mathbf{x}$) in Eq. (21)) was reduced, through the optimization and from prior parameter values, by 31% for NVPs, 64% for shrubs and 54% for boreal C3 grasses and the parameters were optimized within their physical range of variation (see values in Table 2, Table 4 and Table 5). All results that are discussed below were obtained with the optimized parameters set.

Figure 6 shows scatter plots of modelled versus observed living biomass and NPP for the new PFTs. The observations are grouped by bioclimatic zones, including forest-steppe in the south, different taiga ecosystems (south, middle and north), forest-tundra and tundra in the far north. For NVPs the model mean across all sites for biomass and NPP is close to the observed mean (see values in Fig. 6), but the cross-site spread is not well captured. In particular the model spread is too small, especially for the forest-steppe ecosystem, indicating that

the current model structure cannot simulate the spatial variability that is observed between sites. Note also that for the forest-steppe region NPP and living biomass of NVPs are largely under-estimated, by more than 50 and 100 $gC.m^{-2}$, respectively. For cold climate C3 grasses the model spread is much smaller than the observation spread (for both NPP and biomass), although the model mean across all sites is relatively close to the observed value. In particular the model fails to represent the large NPP and biomass for the southern ecosystem (the

forest-steppe), while for the other ecosystems it overestimates the NPP and slightly the biomass. For shrubs the results are relatively similar with also a too low model productivity for the forest-steppe ecosystem. Overall the model captures for each new PFT the mean across all observations but with a large bias for the southern bioclimatic region, where the low simulated values are probably due to a too large water stress in the model (possibly induced by the forcing file at 2° resolution, unable to reproduce local conditions).

We now compare the latitudinal gradient of NPP and biomass over the Central Siberian region shown in Fig. 4. Figure 7 displays model transects from 50°N to 74°N, with mean values calculated over the 78°E to 82°E longitudinal band and over the period 2004-2013, together with the observations aggregated by site (averaged for all year) for each new PFT.

The simulated NPP shows broadly a maximum between 57°N and 65°N for the three PFTs, with a decrease

south of 57°N (by more than a factor two from 57°N to 55°N) and a more progressive decrease north of 65°N. For the NVPs the northern NPP decrease occurs only after 69°N. The observed values are broadly consistent within their uncertainties with the simulated latitudinal gradients for the selected region, although in absence of any observations north of 66°N for shrubs and boreal C3 grasses it is not possible to evaluate the slope of the northern decrease of the simulated productivity. For boreal C3 grasses, if we exclude two sites at 55°N and

67°N having much larger NPP, the other sites reveal a latitudinal pattern similar to the model one, although with smaller values. The simulated total living biomass follows similar latitudinal patterns for the three PFTs, with nevertheless higher biomass for shrubs between 57°N and 65°N due to wood accumulation. The biomass observations for NVPs display the same pattern than the model. For cold climate C3 grasses, even without considering the two sites with very large NPP, the observed living biomass is higher than the model ones

despite the observed lower NPP (Fig. 7-left). It is probably due to large fraction of below-ground biomass of



grasses. For shrubs, the model displays a maximum of biomass around 60°N for this region with large decrease at lower or higher latitudes, that is not directly supported by the set of available observations.

Overall, if the decrease of biomass productivity in the north can be explained by a decline of photosynthesis (due to more extreme conditions), the low value simulated south of 55°N can be attributed to water limitations

(snowfall and rainfall are reduced by 30% in the region 50°N - 55°N compared to 60°N - 65°N), due to change of geographical (or bio-climatic) conditions. Note that two grassland sites that are very closely (65.8°N, 75.4°E and 65.9°N, 75.0°E) have very different NPP (750 $gC.m^{-2}$ and 187 $gC.m^{-2}$) and living biomass values (962 $gC.m^{-2}$ and 260 $gC.m^{-2}$), which illustrate the small-scale variability reported above that cannot be captured by the model.

**3.2    Carbon fluxes and stocks of the new PFTs: spatiotemporal variations**

We now analyse the carbon fluxes (the NPP) and the carbon stocks (July, August and September mean living biomass) obtained with a simulation over the whole boreal zone with the new PFTs (16 PFTs, referred as ORC16; see Fig. 5). The results are averaged over North America (-180°E to -60°E, without Greenland), Europe (-20°E to 40°E) and North Asia (40°E to 180°E) (in Figs. 8-10 and Fig. S1) and we only show the new

PFTs (i.e., boreal C3 grasses, NVPs and shrubs) and the boreal broad leaf summergreen trees (from which shrubs are derived) results (expressed by square meter of each PFT).

**Latitudinal gradients:**

Figure 8 displays latitudinal transects of NPP and living biomass between 45°N and 82°N for each region (see Fig. S1 for the biomass of boreal broadleaved trees). On average we obtain a similar latitudinal gradient in

terms of productivity and biomass for all PFTs, with roughly a maximum in North America around 52°N (with above a continuous decrease until 72°N) and in Asia around 58°N (with a decrease until 78°N) and with a plateau in Europe between 50°N and 70°N (follow by an abrupt decrease). The shape of these latitudinal gradients is primarily controlled by the climate, especially the precipitation and temperature gradients with a strong influence of the topography. For example in Asia the precipitation gradient increases from 45°N (less

than 1 $mm.d^{-1}$) to a maximum around 55°N - 60°N (1.5 $mm.d^{-1}$) and then decreases again northward, while the mean air temperature (at 2m) decreases gradually from 45°N (around +7°C) to 75°N (-13°C). For this region the decrease of precipitation from 60°N to 45°N explains the decrease of NPP and biomass. In Europe the climatic conditions are on average more favourable (e.g. +5°C at 45°N to +10°C à 70°N) which explains the higher productivity and biomasses at high latitude (i.e. around 70°N).

Boreal C3 grasses have on average comparable living biomass but lower NPP than temperate C3 grasses in the southern latitudes where both PFTs are present. On the other hand NVPs always have a much lower productivity and living biomass than grasses (<50% lower). Despite the fact that the NVPs implementation is based on C3 grasses, we also notice that the latitudinal gradients of both productivity and living biomass differ between these two PFTs with smoother latitudinal variations for the NVPs than the boreal C3 grasses,

illustrating also the importance of the added processes for the NVPs (resistances to extreme conditions, see Sect. 2.2.3 and 2.2.4). Similarly, shrubs systematically display a lower NPP (factor two) and much lower biomass (factor 20, Fig. S1) than the corresponding boreal deciduous summergreen trees although with similar latitudinal patterns. The reduced biomass accumulation for shrubs is controlled by the new allometry relations



described in Sect. 2.3.1, a lower residence time (i.e. higher mortality) and a higher fraction of GPP lost as growth respiration (Sect. 2.3.5).

**Temporal evolution:**

Figure 9 shows the yearly time series from 1901 to present day for both NPP and living biomass, averaged north of 55°N, to illustrate the response of the vegetation to climate change. The simulated productivity increases on average for the three regions from 1950 to 2013 (Fig 9.a) by around 25% for boreal C3 grasses, 190% for NVPs and 80% for boreal shrubs (versus 35% for trees). The simulated biomass increases (Fig. 9.b or Fig. S1 with boreal trees) by the same proportion than the NPP for cold climate grasses and NVPs

(respectively +25% and +200%), while for shrubs the increase is stronger (+140%). Note also that the biomass increase for shrubs is much larger than for boreal broad-leaved trees (+20%). Globally, the increase of both NPP and biomass over the last 60 years is substantial for all PFTs, but largest for non-vascular plants and shrubs (see number above), which are more sensitive to climate change and $CO_2$ increase in the model. Note that for shrubs, climate change at high northern latitudes has a direct impact on mortality in winter (Sect.

2.3.3): an increase of the minimum temperature implies a lower mortality. The combination of lower mortality and higher photosynthesis (due to temperature) in Europe, where the temperatures are substantially larger (up to +10°C compared to the other regions), explains the higher increase in simulated biomass and NPP. Note that because the model spin-up was done with climate forcing randomly taken from the period 1901 - 1950 (Sect. 2.6.2) we expect that the impact of climate change in the transient simulation would be small before 1950.

Figure 10 displays the mean seasonal cycle of NPP for the three continental regions (mean over 2004-2013 and above 55°N). As expected, the growing season starts late spring with a sharp increase of the NPP up to July and then a slower decrease up to November, for all PFTs. The seasonality is slightly different for NVPs, for which the maximum is reached earlier (in June), with a small decrease over the summer (with sometimes locally a summer minimum in August) before the large decrease from September on. Such difference is due to

the impact of desiccation during summer time (due to an increase of the water stress, see Sect. 2.2.4) that decreases the maximum potential photosynthesis rate. Finally only small differences in the timing of NPP occur between the three regions, with an earlier start in Europe (1 month), probably due to higher temperatures. Note also that NPP starts slightly earlier in spring for NVPs than for the other boreal PFTs, especially in Europe (Fig. 10). Moreover, the impact of the global increase in temperature is large in spring and autumn,

causing a lengthening of the boreal growing season. The vegetation that could make the best use of these temperature increases may thus get a larger benefit of climate change. This is the case for NVPs, which display an earlier start of the growing season in spring (from March in Europe or April elsewhere) and a later end of season in autumn (after October) (not shown). During these two periods, more than 20% of the annual increase in NPP (Fig. 9) for NVPs occurs, while there is almost no increase for other PFTs.

**3.3    Evaluation of the simulated Leaf Area Index**

Figure 11 displays the mean (over 2004-2013) boreal distributions of Leaf Area Index (LAI) in summer (July, August and September) simulated by ORCHIDEE with the new PFTs (ORC16) and from the GLASS LAI product (see Sect. 2.6.2). It also displays the differences between the simulated LAI (either with the new PFTs description, ORC16, or the old standard description, ORC13) and the GLASS product. Overall the main spatial

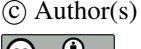



patterns simulated with ORC16 match relatively well the patterns of the GLASS product with i) a latitudinal band with higher LAI around 60°N in Eurasia and below 60°N in northern America and ii) lower LAI at low latitudes in central Siberia and in above 65°N in Siberia and North America. However, too low LAI seems to be simulated in western Siberia. Comparison between GLASS product and the two model simulations (ORC16 and ORC13) indicates an overall improvement of the simulated LAI with the inclusion of the new boreal PFTs. A substantial decrease of LAI in Northern Europe (from 55°N), Northern-Western Siberia (from 55°N and until 135°E) and Northern America (from 50°N) is simulated in ORC16 compare to ORC13, which is in better accordance with GLASS product. These lower values in ORC16 are attributed, north of 65°N in Asia and America, to the introduction of NVPs in replacement of C3 grasses (Sect. 2.5.2) with lower LAI (see Sect. 3.2). In addition, the introduction of cold climate C3 grasses and shrubs with lower maximum LAI (e.g. 2.5 for shrubs against around 4 for tree PFTs) also contributes. Elsewhere, ORC16 and ORC13 simulations present on average similar LAI anomalies (mainly located in the south), except for Alaska and Eastern Siberia where ORC16 – GLASS anomalies are slightly more negative than with ORC13.

### 3.4    Biophysical impacts of the new boreal vegetation description

We now investigate the impacts of the new vegetation types on a few key variables related to the energy and water budgets (Fig. 12). The annual albedo shows a significant increase (up to 0.1) with the new boreal PFTs (ORC16) compared to the standard version (ORC13). The higher albedo occurs primarily in winter and early spring (see January and April in Fig. 12) in northern high latitudes (North of 60°N), whereas there is nearly no change during summertime and early autumn. If we consider the contribution from vegetation only (i.e. the mean albedo of the fraction of the grid covered by vegetation without the effect of snow cover and without bare soil) there is a small decrease with the new PFTs in most regions, except in northern-central Siberia. These changes are due to the LAI of the different PFTs that control the fraction of the grid effectively covered by the vegetation foliage. The higher vegetation albedo in ORC13 can be attributed to the larger values of the LAI for trees compared to shrubs and for temperate C3 grasses compared to cold climate C3 grasses. In the Siberian region, the lower vegetation albedo in ORC13 occur in early spring, while higher values are present all year-round, due to changes in LAI with NVPs. Note that changing from a C3 deciduous grassland to an evergreen PFTs (i.e. the NVPs) impact the albedo even in winter time if the snow cover is not complete. Overall, the small changes of vegetation albedo and its dissymmetry with the changes in total albedo indicate that the substantial increase in the total albedo is linked to changes in the snow albedo and/or snow cover. The snow cover is controlled by the snow depth, the vegetation type and its roughness (see Sect. 2.3.4).

Roughness length is stable throughout the year and clearly decreases with the new vegetation types (up to -0.5 m, which represents at least a decrease of 25%, Fig. 12), due to height differences between trees and shrubs, the height being used to compute the roughness length (Eq. (17)). Contrariwise, the snow depth and albedo are not impacted by vegetation changes, because there is no difference between trees and shrubs concerning the snow compaction (described in Sect. 2.3.2). Given that roughness and snow depth contribute to the albedo through the fraction of snow on the vegetation (Eq. (18)), the modification of winter albedo is due mostly to roughness length changes.





As expected, transpiration is mainly affected during the summer period with much lower values (up to -0.5 mm.d$^{-1}$) in July around 60°N in West Eurasia and below 60°N in North America in the ORC16 simulation versus the ORC13 simulation. Crossing this information with the vegetation map, this is probably due to the replacement of trees by shrubs; shrubs have a lower leaf biomass, a lower photosynthesis rate (Figs. 8-10,12),

and a lower roughness (Fig. 12, inducing less turbulent flow) leading to a lower transpiration. On the other hand, the introduction of NVPs, which have a higher stomatal conductance that could lead to an increase in transpiration, does not seem to have a major impact. However, if we focus on land surfaces North of 65°N (representing 11.2 millions km$^2$), the inclusion of the new PFTs slightly changes the components of the water budget. The inputs are identical between both simulations and the snowfall represents 53% of the total annual

precipitation. The outputs represent respectively for ORC16 and ORC13 0.22 and 0.21 mm.d$^{-1}$.m$^{-2}$ for the runoff, 0.11 and 0.08 mm.d$^{-1}$.m$^{-2}$ for the drainage, 0.54 and 0.58 mm.d$^{-1}$.m$^{-2}$ for the evaporation, and 0.17 and 0.19 mm.d$^{-1}$.m$^{-2}$ for the sublimation. There is thus a slight decrease of evaporation (-6%) and sublimation (-11%) with the new boreal vegetation description, compensated for by an increase of the runoff (+4%) and drainage (+27%). The lower transpiration in summer simulated by ORC16 (up to 5mm.d$^{-1}$, see Fig. 12) is less

substantial during other seasons, and it could be partly compensated by bare soil evapotranspiration. Finally, the global water balance leads to an increase of runoff and drainage to 135 km$^3$.y$^{-1}$ (+10%) north of 65°N.

We finally investigate the impact of the new PFT description on the soil energy budget and more specifically the potential impact on the future reduction of the permafrost areas. Figure 13.a represents the thickness of the active layer, which corresponds to the maximum depth of the 0°C isotherm. The model represents the

permanent frozen soil North of 50°N in North America and East Asia and North of 60°N elsewhere. Figure 13.b displays the change in active layer thickness with the new PFTs (ORC16). At its southern limit, the active layer thickness seems to increase on average and by up to 1 m in ORC16 compared to ORC13. To determine the role of each vegetation type, differences in the profiles of the annual soil temperature (mean over 2004-2013) are displayed in Fig. 13.c for three locations with different vegetation coverage. The profile at 169°E

63°N, selected for its high NVP coverage (40%), shows colder soil temperatures in the ORC16 simulation (-0,15 °C on average from the surface to 16 m), with warmest surface (0 to 1 m) temperature in winter (up to +0,25°C) and coldest surface temperature in summer (up to -0.7°C). This result indicates a lower surface conductivity, due to the insulation of the first centimeters of soil by NVPs (see Sect. 2.2.5). The 45°E 63°N profile was selected because of large differences between the ORC16 and ORC13 active layer thicknesses. It

shows a higher soil temperature in the ORC16 simulation (+0.18 °C on average, with low seasonal variation) and corresponds to a low coverage by NVPs (3%). This higher temperature can be explained by a large fraction of the new shrubs and C3 cold climate grasses (> 50%) inducing a lower transpiration (Fig. 12). The reduction of transpiration in ORC16 leads in turn to a higher soil humidity and thus a higher thermal conductivity (see $C_{wet}$ and $C_{dry}$ values in Table 2). Finally, the 65°E 61°N profile was selected in some point where no active

layer differences was noted. It includes 75% of new boreal PFTs from which 14% of NVPs and displays colder soil temperature in ORC16 up to 5 meters (although varying with depth), but similar temperature between ORC16 and OCR13 deeper into the soil (differences below 0.05°C on average).

Overall, the impact of the thermal insulation by NVPs seems to be compensated by an increase of soil humidity brought by the boreal PFTs. The active layer becomes deeper with the new boreal vascular plants (boreal C3

grasses and shrubs) due to higher soil conductivity, while the presence of NVPs decreases the active layer





thickness with higher soil insulation. The coverage differences between NVPs and new vascular plant explains the global positive difference values in Fig 13.b.

## 4       Summary and conclusions

### 4.1       Challenges associated to the description of new boreal vegetation

In this study we added non-vascular plants, boreal shrubs and boreal C3 grasses in the land surface scheme ORCHIDEE. While the implementation of boreal C3 grasses boils down to parameter changes (see Table 5), new key processes have been introduced for the other two PFTs:

- For shrubs, a new allometry was defined (compared to trees) in order to simulate a realistic vegetation height, which is further used to describe shrubs interactions with snow (Sect. 2.3).

- For NVPs, we opted for an "indirect" representation of their physiological functioning using the same process-representation as for vascular plants but with specific modifications (parameters and equations). A shallow root profile was chosen to represent the access to surface water. A large leaf water and $CO_2$ conductance was introduced to represent the lack of stomata. Additionally, a specific plant resistance to water stress, the impact of a NVPs on soil thermal properties and a modification of

litter decomposition were implemented (Sect. 2.2).

In order to calibrate the main parameters of these new boreal PFTs, observations of net primary productivity and living biomass from Siberia were used (Sect. 2.5.1) with a standard Bayesian optimization procedure (see Sect. 2.6.1). Note that the large data spread (Figs. 6-7) due to large spatial variability at the scale of a few meters could not be represented by the model with a 2° climate forcing and no explicit representation of the

underground vegetation (and competition) and edaphic conditions. Note that the better adequacy between the observations and the simulation for NVPs is partly due to more homogenous data.

Given the limitations discussed in the sections above, we suggest new developments to improve the realism of the simulated water, carbon and energy fluxes for the arctic region. First, it would be important to better represent the spatial heterogeneity of edaphic conditions, possibly with the use of topography information (i.e.,

to improve the water stress computation), and the vertical structure of the vegetation in coherence with light penetration and intra-canopy gradients of climate variables, as in Ryder et al. (2016). A more accurate vertical representation of the vegetation structure implies to introduce vegetation strata with the possibility to have under-storey vegetation, such as shrubs, grasses or NVPs under a tree canopy (e.g., in Frolking et al., 1996). Furthermore, it could be important to take into account the impact of other chemical components and

processes, such as the availably of oxygen in the upper soil to represent anoxic conditions and of nitrogen to account for possible limitation on plant productivity (Epstein et al., 2000; Bond-Lamberty and Gower, 2007; Goll et al., 2012; Koven et al., 2013), especially for NVPs. Thereby, extreme conditions would be more realistically simulated (such as for peatlands) avoiding the use of proxies for key environmental drivers (such as soil humidity for anoxic conditions). Concerning shrubs, we selected a boreal broad-leaved summergreen

phenology, although in reality there is a mix of summergreen and evergreen needled-leaved shrubs. Given that the main changes introduced for the shrub PFT are linked to the allometry and the interaction with snow (Sect. 2.3), it should be straightforward to split this PFT into different types, as already done for trees (Table 1), with only a few varying parameters (such as phenology type, minimal critical temperature or $Vc_{max(25)}$). For other



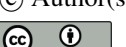

climatic regions than the Arctic, new processes may need to be added, such as root development for shrubs in savannahs. In a similar way we can split NVPs between lichens and mosses. Furthermore, to improve the dynamic of shrubs-snow interactions, it would be important to implement an energy balance and a snow mass balance for each PFT, separately. Thereby, the interactions between wind, snow deposition and compaction

and vegetation structure could be integrated (McFadden et al., 2001). Finally, the implementation of other processes such as soil flooding (due to permafrost thawing for example) should be also considered as a crucial additional step.

### 4.2    Biogeochemical impacts of the new boreal vegetation

The overall biogeochemical behaviour of the new boreal PFTs is significantly different than that of the original
PFTs. Cold climate grasses exhibit a lower productivity than the original C3 grasses because of their lower maximum rate of carboxylation ($Vc_{max(25)}$, in Table 5), but a comparable biomass. NVPs globally have a lower productivity and biomass than temperate and boreal C3 grasses (Figs. 8-9), which is also explained by the low $Vc_{max(25)}$ (respectively 70, 40 and 28 $\mu mol.m^{-2}.s^{-1}$). However, these lower mean values mask a better adaptation of NVPs to the northern latitudes, with higher productivity in spring and at the end of autumn (Fig. 10) and a
decline in summer due to a water stress. Such adaptation arises from few specific processes implemented for the NVPs such as the resistance to desiccation or the adapted turnover, stomatal conductance and photosynthesis capacity. Shrubs also have a lower productivity and a much lower biomass than trees (Figs. 8-9) because of their lower LAI, new plant allometry and adapted mortality and respiration. Shrubs have an increased mortality induced by cold temperatures, but they are on the same time protected by snow (thermal
protection; Eq. (15)). On the other hand, trees do not have this increased mortality with extreme temperature and it could be beneficial to include this effect when the vegetation is fixed, using for instance Eq. (14) that is only applied in ORCHIDEE when the vegetation cover is dynamicaly calculated (Krinner et al., 2005; Zhu et al., 2015).

Spatially, the northern limit of shrubs is situated further south than the northern limit of NVPs and cold climate
grasses, as described in CAVM Mapping Team et al. (2003) and Loveland et al. (2000). Moreover, there are differences between the three boreal regions (North America, North Asia and Europe) due to climatic conditions: productivity and total living biomass decrease rapidly with latitude in Northern America, more slowly in Asia, while in Europe they remain at a high level far north (Fig. 8). Overall for the arctic regions, the total carbon flux is dominated by the prescribed vegetation distribution and more specifically by the fractions
of trees and temperate grasses (Fig. 5). The inclusion of new boreal vegetation types decreases considerably the productivity, the total living biomass, and thus the LAI, which becomes more closer to satellite observations (GLASS product, Fig. 11) (Liang et al., 2013; Xiao et al., 2014). This implies that in previous simulations (and in particular those for the last IPCC report), considering vegetation without boreal shrubs and grasses might have induced a significant overestimation of biomass and productivity in northern latitudes.

As expected, the global increase of NPP, GPP and biomass over the last 60 years (Fig. 9) reveals the vegetation response to global warming and increased $CO_2$. This response is substantial, especially for NVPs and boreal shrubs and particularly for the accumulation of biomass. Thus in boreal regions the new PFTs are more sensitive to climate change than the original ones, implying that the standard ORCHIDEE version under-estimates the potential changes of vegetation biomass and productivity. In addition, shifts of vegetation are





already observed (Frost and Epstein, 2014; Zhu et al., 2016) and must be taken into account in dynamical vegetation modelling.

Based on this study, we foresee several applications for the biogeochemical cycles. First, it is crucial to update the dynamic vegetation module of ORCHIDEE in order to account for and to calibrate the competition
between all PFTs. This requires defining for instance the drivers of the competition between grasses and NVPs and between shrubs and trees. Such developments will open the road for new studies of boreal vegetation changes, in the future or in the past, in liaison with climate changes. Second, the simulation of more realistic NPP and biomass in boreal landscapes could help to better simulate the dynamic of past boreal vegetation cover and boreal carbon stocks. For example, for the Last Glacial Period, it would enable a better estimation of
carbon accumulation in the soil and thus of carbon stocks present in today's permafrost.

### 4.3     Biophysical impacts of the new boreal vegetation

As illustrated in the results section (Figs. 12-13), multiple impacts on the energy and water balance of boreal ecosystem occur with implementation of new PFTs in the ORCHIDEE model.

The changes in vegetation albedo result directly from changes in vegetation cover: in this study the vegetation
map is prescribed and PFT-dependent albedo parameters are identical for cold climate grasses and NVPs / Shrubs and the corresponding standard PFTs (grasses / trees). Therefore, with its lower LAI, the new boreal vegetation induces a lower vegetation albedo (without snow cover), except in winter for areas where newly introduced evergreen NVPs are present. In contrast, the overall albedo increase (Fig. 12) does not seem directly impacted by the vegetation distribution. This depends on a combination of the local high vegetation albedo due
to NVPs, and the decrease of roughness length, due to the substitution of a fraction of trees by shrubs (Sect. 2.5.2), which implies an increase of snow cover fraction (Eq. (18)).

The substitution of a fraction of trees by shrubs largely contributes to the summer transpiration decrease. The active layer thickness (Fig. 13) and permafrost extension are impacted by the NVPs through two competing effects. NVPs insulate the soil but also increase the soil thermal conductivity through an increase of soil
humidity due to a global decrease of transpiration. Overall, we obtain a weak or negative impact of the new boreal vegetation implementation on the permafrost extent. This is at odds with results reported elsewhere (Soudzilovskaia et al., 2013; Chadburn et al., 2015). Further investigations are required to determine whether this is an artefact of our choice to replace the standard soil thermal capacity and conductivity by intermediate values between those from NVPs and mineral soil. One option would be to treat the NVPs as a layer with its
own energy budget and thermal characteristic above the soil. Also note that, while the NVP heat conductivity and heat capacity used in this study are in accordance with other experiments (Soudzilovskaia et al., 2013; Chadburn et al., 2015), the average thickness of mosses in our simulation is lower than the one used in Chadburn et al. (2015), where it was fixed. Moreover, NVPs have an impact on the soil water dynamic, not well represented in ORCHIDEE. For example, in JULES Chadburn et al. (2015) chose to use a suction
equation from Brooks and Corey (1964) to compute the plant water uptake and represent the "spongy" effect of NVPs. In ORC-HL-VEGv1.0, three options were therefore considered: (1) increase the leaf interception and infiltrate part of this water into the soil, (2) limit the runoff in order to hold more water on the upper soil layers, or (3) increase the water retention by changing soil parameters controlling diffusion and drainage. However, given that in the current version of ORCHIDEE a unique soil water budget is performed for the entire





herbaceous layer (no distinction between NVPs and grasses), it was not possible to represent the suction effect of NVPs more precisely. The water content of surface layers is thus probably underestimated and can impact the soil conductivity.

Overall, the total runoff and drainage above 65°N with the new vegetation increases by 11% with respect to the

13-PFT case, and reach around 140km$^{-3}$.y$^{-1}$ (see Sect. 3.4). Future replacement of NVPs and grasses by shrubs and trees could therefore counteract the direct effect of atmospheric $CO_2$ increase (i.e. decrease of transpiration) on Arctic river runoff (e.g. Gedney et al., 2006).

In this study we improved the description of boreal biophysical processes, but we did not consider the

feedbacks between vegetation and climate. For example, the simulated increase of albedo, with the new boreal PFTs and new albedo formulation (Sect. 2.3.4), could reduce locally the surface air temperature and potentially impact the snow dynamic for instance. Moreover, the decrease of roughness length, due to the replacement of trees by shrubs (Sect. 2.3.1), will impact the exchange of momentum between the surface and the atmosphere and thus likely impact regional to large scale circulation patterns (e.g., Vautard et al., 2010). It is thus

necessary to evaluate all potential feedbacks between vegetation and climate with such improved description of boreal vegetation in the IPSL-CM earth system model (ORCHIDEE being the surface component).

**Code availability**

The code and the run environment of ORCHIDEE are open source (http://forge.ipsl.jussieu.fr/orchidee). Nevertheless readers interested in running the ORC-HL-VEGv1.0 version described in this paper can have

access to the code (available at https://github.com/ArseneD/ORC-HL-VEG commit b74ae16) and are encouraged to contact the corresponding author for full details and practicality.

**Acknowledgements**

This study was made possible thanks to the GAP Swedish-French project and Page21. We would also like to thank Deborah Verfaillie for her help in correcting the language.



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



| Bare soil | | | |
|---|---|---|---|
| Trees | Tropical | Broadleaf | Evergreen |
| | | Broadleaf | Raingreen |
| | Temperate | Needleleaf | Evergreen |
| | | Broadleaf | Evergreen |
| | | Broadleaf | Summergreen |
| | Boreal | Needleleaf | Evergreen |
| | | Broadleaf | Summergreen |
| | | Needleleaf | Summergreen |
| *Shrubs | *Boreal | *Broadleaf | *Summergreen |
| Grasses | Natural | C3 | Global |
| | | | *Arctic |
| | | C4 | |
| | Crops | C3 | |
| | | C4 | |
| *Non-Vascular (C3) plants | | | |

**Table 1: PFTs included in ORCHIDEE. New PFTs incorporated in this study are indicated with asterisks.**





| Parameters | Description | Original C3 grasses | Non-Vascular Plants |
|---|---|---|---|
| Phenotype | | Summergreen | Evergreen |
| Organs | Organs proportion | roots, reserves, leaves, fruits (10%) | Leaves (95%), fruits (5%) |
| $g_0$ (mol.m$^{-2}$.s$^{-1}$.bar$^{-1}$) | Stomatal conductance when irradiance is null | 0.00625 | 0.052 * |
| $a_1$ (-) | Empirical constants | 0.85 (all PFT) | 0.85 |
| $b_1$ (-) | Empirical constants | 0.14 | 0.41 * |
| Senescence (day) | Theoretical number of days before senescence | 120 | 470 * |
| $d_0$ (day) | Delay before increasing the turnover (if NPP≤0) | - | 20 |
| $d_m$ (day) | Number of days when the fraction of biomass loss is maximal (if NPP≤0) | - | 60 |
| $d_f$ (day) | Maximum number of day for this extra turnover (if NPP≤0) | - | 130 |
| $k_{l\,max}$ (days) | Maximal fraction of biomass loss (if NPP≤0) | - | 0.05 * |
| $LAI_{lim}$ (-) | Threshold leaf area index (for turnover) | - | 2.4 * |
| $l_{coef}$ (-) | Coefficient | - | 0.014 * |
| $r_p$ (-) | Parameter to control root profile | 4 | 18 * |
| $w_{s\,min}$ (-) | Minimum hydric stress before any desiccation effect | - | 0.8 |
| $d_{off}$ (-) | Offset of desiccation effect | - | 0.55 * |
| $\rho$ (gC.m$^{-3}$) | Density | - | $0.5 \cdot 10^4$ |
| $C_{dry}$ (J.m$^{-3}$.K$^{-1}$) | Dry soil thermal capacity | 1.80 | $0.29 \cdot 10^6$ |
| $C_{wet}$ (J.m$^{-3}$.K$^{-1}$) | Wet thermal capacity | 3.03 | $4.29 \cdot 10^6$ |
| $C_{ice}$ (J.m$^{-3}$.K$^{-1}$) | Ice thermal capacity | 2.11 | $3.26 \cdot 10^6$ |
| $\lambda_{dry}$ (W.m$^{-2}$.K$^{-1}$) | Dry soil thermal conductivity | 0.4 | 0.092 |
| $\lambda_{sat\_wet}$ (W.m$^{-2}$.K$^{-1}$) | Wet thermal conductivity | 0.6 | 0.754 |
| $\lambda_{sat\_ice}$ (W.m$^{-2}$.K$^{-1}$) | Ice thermal conductivity | 2.2 | 0.715 |
| $m_{c(0)}$ (-) | Constant | | 1.178 |
| $m_{c(1)}$ (-) | Constant | | -1.12 |
| $m_{c(2)}$ (-) | Constant | | 2.22 |
| $m_{c(3)}$ (-) | Constant | | -1.40 |
| $LAI_{max}$ (m$^2$.m$^{-2}$) | Maximum Leaf Area Index | 2 | 3.06 *[&]*** |
| $Vc_{max(25)}$ (µmol.m$^{-2}$.s$^{-1}$) | Maximum rate of carboxylation at 25°C | 70 | 28 * |
| $SLA$ (m$^2$.gC$^{-1}$) | Specific Leaf Area | $2.6 \cdot 10^{-2}$ | $0.84 \cdot 10^{-2}$ * |
| $f_{m\_resp}$ (gC.gC$^{-1}$.day$^{-1}$) | Maintenance respiration coefficient at 0°C | $2.62 \cdot 10^{-3}$ | $2.57 \cdot 10^{-3}$ * |

**Table 2: Non Vascular Plant parameters.**

* Optimized parameter (see Sect. 2.6.1)

** Estimated from Yoshikawa et al. (2002) and O'Donnell et al. (2009)

*** Estimated from Bond-Lamberty and Gower (2007)

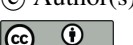



| Parameters | Description | Original values | Ground vegetation (Bare soil, Grasses and NVPs) | High vegetation (Shrubs and Trees) |
|---|---|---|---|---|
| $a_{sc}$ | Snow settling parameter ($s^{-1}$) | $2.8 \times 10^{-6}$ | $1.4 \times 10^{-6}$ | $4.2 \times 10^{-6}$ |
| $b_{sc}$ | Snow settling parameter ($K^{-1}$) | 0.04 | 0.02 | 0.06 |
| $c_{sc}$ | Snow settling parameter ($m^3.kg^{-1}$) | 460 | 230 | 690 |
| $a_\eta$ | Snow Newtonian viscosity parameter ($K^{-1}$) | 0.081 | 0.0405 | 0.12 |
| $b_\eta$ | Snow Newtonian viscosity parameter ($m^3.kg^{-1}$) | 0.018 | 0.009 | 0.027 |
| $\eta_0$ | Snow Newtonian viscosity parameter (Pa.s) | $3.7 \times 10^7$ | $1.85 \times 10^7$ | $5.55 \times 10^7$ |

Table 3: Snow compaction parameters. Original values from Wang et al. (2013) and herbaceous and high vegetation values are choose to stay in the range value proposed by Wang et al. (2013).



| Allometry | | | | |
|---|---|---|---|---|
| **Parameters** | **Description** | **Trees** | **Shrubs** | |
| | | Pipe tune | Pipe tune (like trees) | Aiba and Kohyama (1996)** |
| $A$ | Allometry constant | - | - | 0.75 |
| $\beta$ | Allometry constant | 40.0 | 8.0 | $\text{Log}(\beta) = 2.42$ |
| $\gamma$ | Allometry constant | 0.5 | 0.55 | 1.15 |
| $\alpha$ | Allometry constant | 100.0 | 216.9 | 0.8 |
| $\delta$ | Allometry constant | 1.6 | 1.6 | - |
| $H_{max}$ (m) | Maximum height | 15 | 3.5 * | 3.5 * |
| $H_{f\_dia}$ (0-1) | Maximum height used to compute the diameter | - | - | 0.90 |
| $h_c$ | Minimum height factor | 10 | 10 | 10 |
| Other **Parameters** | | | | |
| **Parameters** | **Description** | **Trees** | **Shrubs** | |
| $k_{ce}$ (-) | Coefficient of mortality due to extreme coldness | 0.04 | 0.04 | |
| $T_{min,crit}$ (°C) | Minimum critical temperature | -45 | -45 | |
| $z_{0\_c}$ (m) | Roughness constant | 16 | 16 | |
| $z_{0\_bs}$ (m) | Roughness of the bare soil | 0.01 | 0.01 | |
| $\Delta_{zo}$ (-) | Width of the transition zone when $d_s$ is around $H_{PFT}$ | 0.3 | 0.3 | |
| $\xi$ (-) | Snow fraction constant | 5 | 5 | |
| $SLA$ (m².gC⁻¹) | Specific Leaf Area | $2.6\ 10^{-2}$ | $2.7\ 10^{-2}$ * | |
| $LAI_{max}$ (m².m⁻²) | Maximum Leaf Area Index | 4.5 | 2.5 * | |
| $Vc_{max(25)}$ (µmol.m⁻².s⁻¹) | Maximum rate of carboxylation at 25°C | 45 | 38 * | |
| Residence Time (y) | | 80 | 32 * | |
| $f_{g\_resp}$ (0-1) | Fraction of GPP which is lost as growth respiration | 0.28 | 0.59 * | |

**Table 4: Shrub parameters.**

* Optimized parameter (see Sect. 2.6.1)

** Adapted from Martínez and López-Portillo, 2003





| Parameters | Description | Original C3 grass | Boreal C3 grass |
|---|---|---|---|
| $Vc_{max(25)}$ (mol.m$^{-2}$.s$^{-1}$) | Maximum rate of carboxylation at 25°C | 70 | 40 * |
| $E_a$ (J.mol$^{-1}$) | Activation energy | 71 513 | 71 513 |
| $E_d$ (J.mol$^{-1}$) | Deactivation energy | 200 000 | 200 000 * |
| $a$ (J.mol$^{-1}$.K$^{-1}$) | Entropy constant | 668.39 | 668.39 |
| $b$ (J.mol$^{-1}$.K$^{-1}$.°C$^{-1}$) | Entropy constant | -1.07 | 0.0 * |
| $J_{max(25)}$ | Maximum rate of electron transport at 25°C | | |
| $E_a$ (J.mol$^{-1}$) | Activation energy | 49 884 | 49 884 |
| $E_d$ (J.mol$^{-1}$) | Deactivation energy | 200 000 | 200 000 * |
| $a$ (J.mol$^{-1}$.K$^{-1}$) | Entropy constant | 659.7 | 659.7 |
| $b$ (J.mol$^{-1}$.K$^{-1}$.°C$^{-1}$) | Entropy constant | -0,75 | 0 * |
| $r_p$ (-) | Parameter to control root profile | 4 | 5.6 * |
| $SLA$ (m$^2$.gC$^{-1}$) | Specific Leaf Area | 2.6 x 10$^{-2}$ | 2.2 x 10$^{-2}$ * |
| $R$ (J.mol$^{-1}$.K$^{-1}$) | Ideal gas constant | 8.314 | 8.314 |

**Table 5: Boreal C3 grasses parameters.**

\* Optimized parameter (see Sect. 2.6.1). Note that $J_{max}$ and $Vc_{max}$ parameters, namely $E_d$ and $b$, were linked for the optimization.



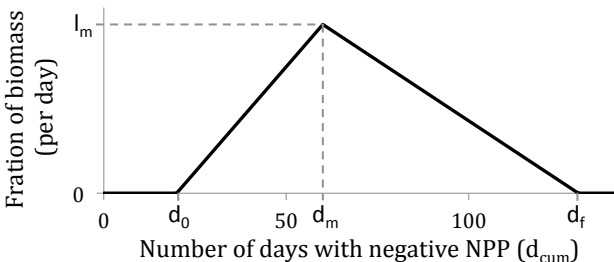

**Figure 1: Additional non-vascular biomass loss turnover rate ($k_l$ in d$^{-1}$) during the non-growing season period when NPP is lower or equal than zero, starting at 0 on the horizontal axis.**





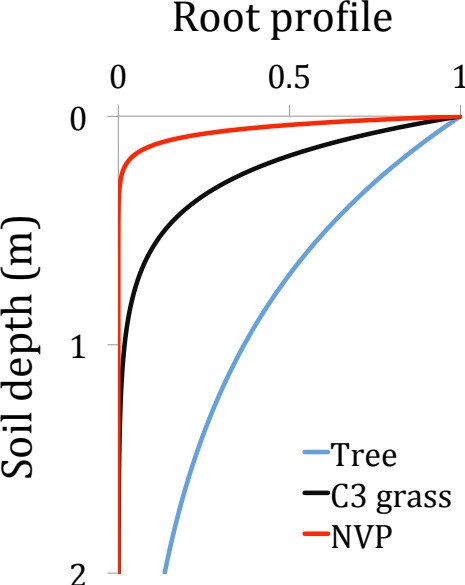

**Figure 2: Root profile of boreal broadleaf trees, C3 grasses and soil water uptake profile for NVPs.**





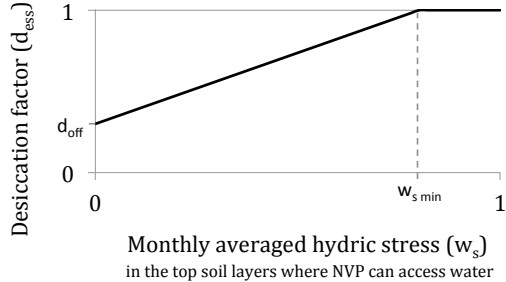

**Figure 3: Desiccation function for non vascular plants.**





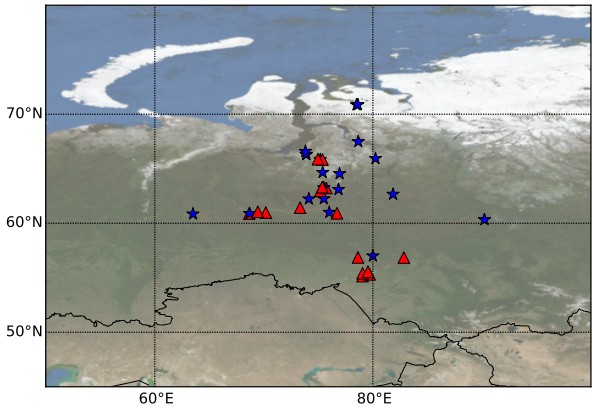

**Figure 4: 36 sites of vegetation green biomass and Net Primary productivity (NPP). Triangles in red: sites with NVPs, grasses and shrubs at the same location, stars in blue: sites with only NVPs.**



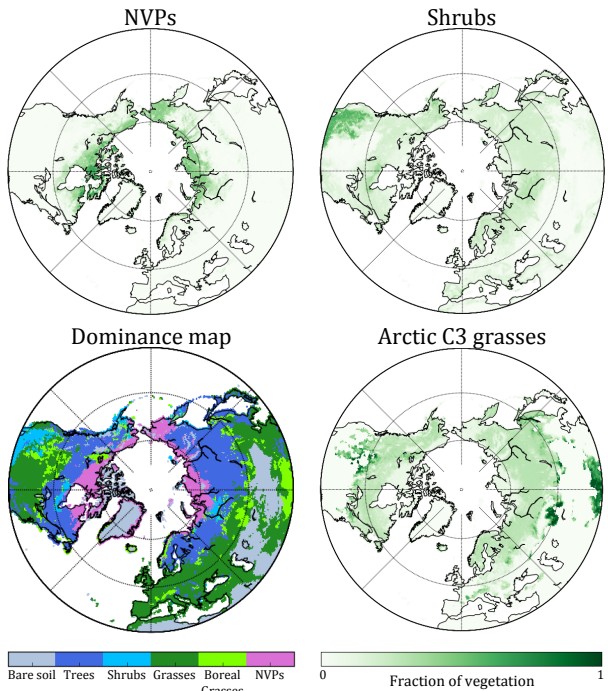

**Figure 5: Map of new PFTs vegetation coverage and dominance.**





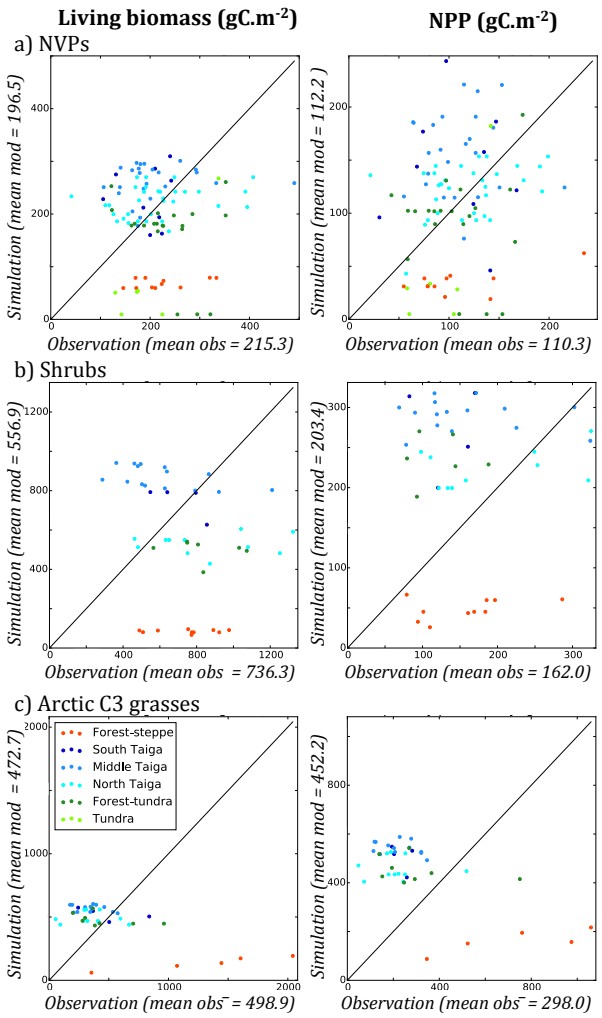

**Figure 6: Model versus observed values for the total living biomass (left column) and the NPP (right column), for NVPs (a), shrubs (b) and cold climate grasses (c). The mean values across all sites and all years are displayed for the model and the observations. The colour indicates the associated bioclimatic zones.**

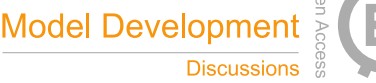



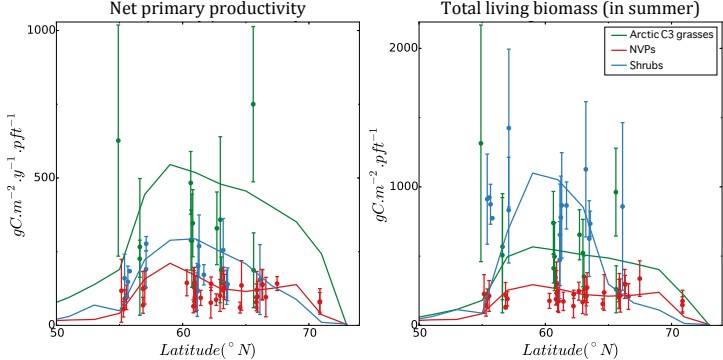

**Figure 7: Latitudinal transects of the modelled and observed annual Net Primary Productivity and total living biomass in summer (July, August and September) for the new PFTs, namely boreal C3 grasses, non-vascular plants and shrubs. The simulated values are averaged over the longitudinal band 78°E - 82°E, and per latitudinal bands of 2 degrees, starting at 50°N.**





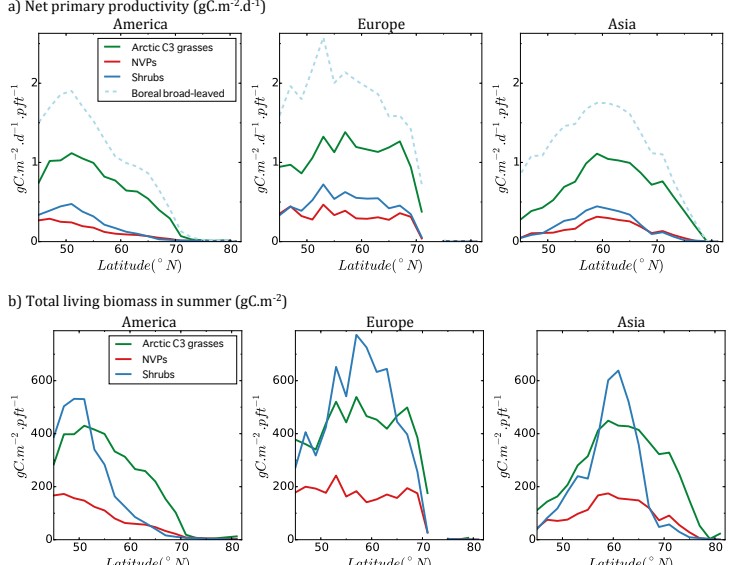

**Figure 8: Latitudinal transects of the mean 2004-2013 net primary productivity (NPP) (a) and total living biomasses (b) of new PFTs (boreal C3 grasses, NVPs and boreal shrubs) and boreal broad-leaved tree (dashed, only in a), simulated in ORC16.**





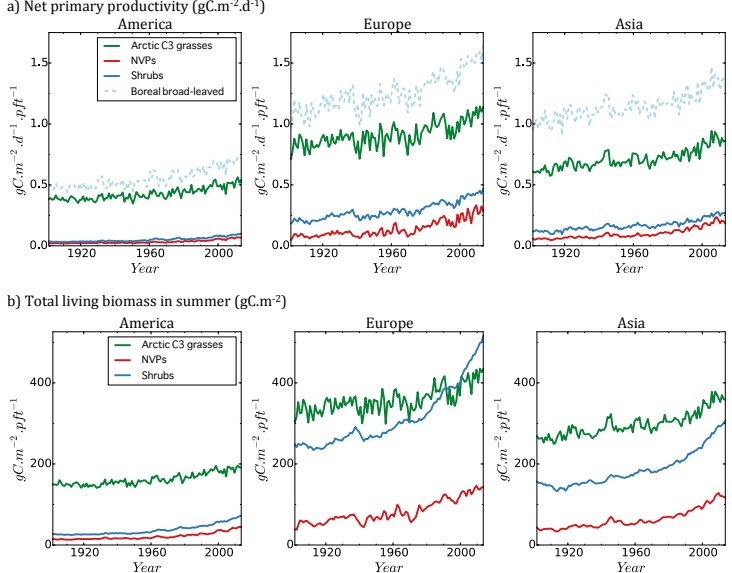

**Figure 9: Time series from 1901 to 2013 and from 55°N of Net Primary Productivity (a) and total living biomass (b) of new PFTs (boreal C3 grasses, NVPs and boreal shrubs) and boreal broad-leaved tree (dashed, only in a).**

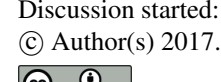



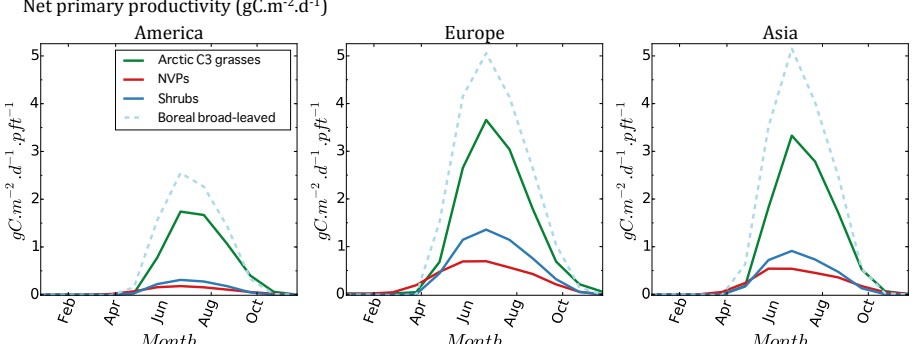

**Figure 10: Inter-annual net primary productivity time series (mean 2004-2013) of new PFTs (boreal C3 grasses, NVPs and boreal shrubs) and boreal broad-leaved tree (dashed).**





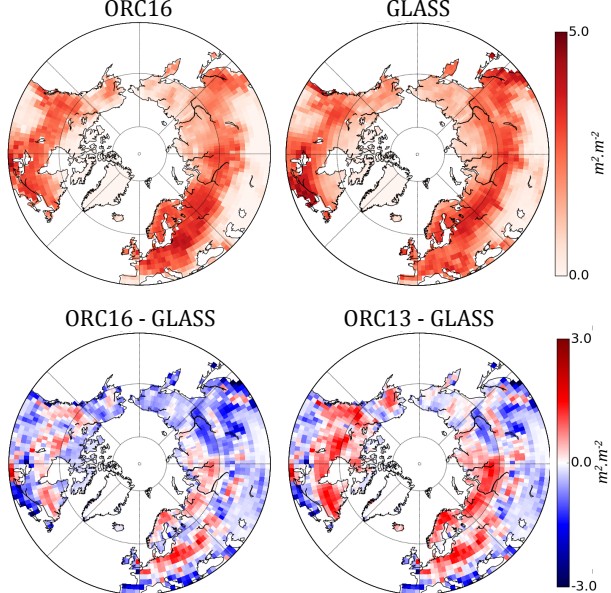

**Figure 11: Global maps of leaf area index (LAI) in summer (mean of July, August and September between 2004 and 2013) simulated with the new PFTs (ORC16) and derived from satellite observations (GLASS product), as well as the difference between the simulation with the new PFTs or the old 13 PFTs (ORC16 and ORC13 respectively) and the GLASS product.**





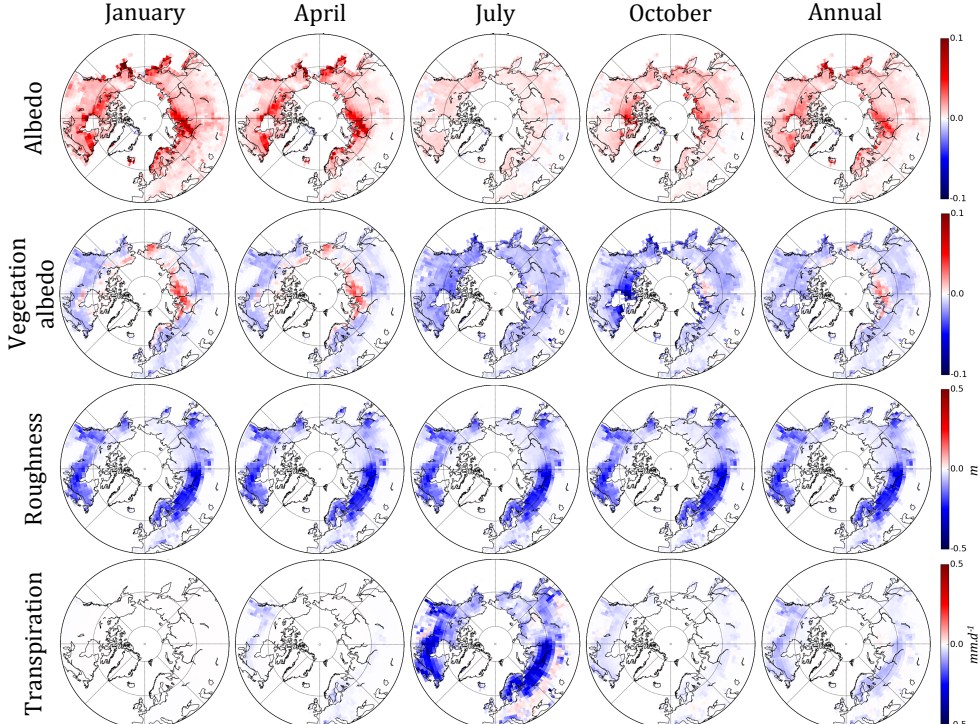

**Figure 12: Maps of the differences between the simulation with 16 PFTs (ORC16 with new boreal PFTs) and the simulation with the 13 PFTs (ORC13 standard version), for albedo (total albedo and vegetation only without snow and bare soil contribution), roughness and transpiration for January, April, July, October, and the annual mean (mean over the period 2004 to 2013).**

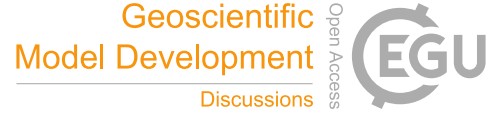



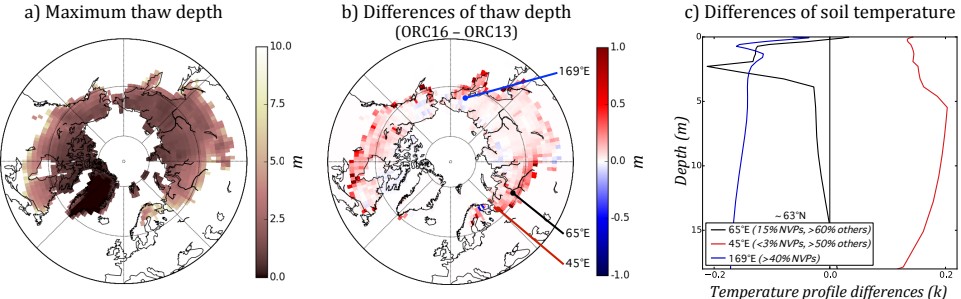

**Figure 13: Map of a) the maximum thaw depth (i.e., the active layer thickness) for the simulation with 16 PFTs (ORC16); b) differences between ORC16 and the simulation with the 13 PFTs (ORC13) and c) soil temperature profile differences at three selected points (63°N and 45°E, 65°E and 169°E) between ORC16 and ORC13.**

