# Peer review of "Towards a more detailed representation of high-latitude vegetation in the global land surface model ORCHIDEE (ORC-HL-VEGv1.0)"

_Geoscientific Model Development, 2017_

## Referee Comment (RC1) · Anonymous Referee #1 · 3 May 2017

This paper represents a great amount of work in model development, and in general it is well justified, well written, and the availability of such a model will contribute towards science both through using the improved model and informing other model developers. Therefore I recommend that it should be published in this journal, but with some clarifications and a bit of consolidation.

Firstly, the paper is rather long. I am not convinced that separating the analysis in figures 8-10 into different continents (Europe/Asia/America) is really relevant to the model developments here. Differences between the PFT's should still be visible in the aggregate results. Consolidating these would reduce the figures and you could remove some of the discussion of inter-continental differences from the text. These

are interesting but the paper would benefit from being a bit shorter.

Throughout the manuscript you have used the word "summergreen", which I have never heard before and we always use "deciduous". I'm not sure summergreen in really a word in English and maybe you should used deciduous instead? Sorry if I'm wrong here.

Specific comments * P1 Line 17 what you mean by "a larger phenological plasticity" isn't entirely clear to me. Maybe because I am not a specialist in vegetation but I think this will be read by other 'general' land surface modellers so could maybe be a bit clearer. Do you mean the phenology varies more in the season? Or more quickly over time?

* P1 Lines 23-26. Please check all of these numbers for the percentage changes. I can't find them all in the main text or they don't seem to be consistent - for example, the change in roughness is quoted as 25% in the main text (page 20, line 33), but 41% in the abstract.

* P5 line 6 "coefficients a1 and a2" - should be "b1" instead of "a2" as it seems to be called b1 in the table. Furthermore, you said you chose values so that stomatal conductance would not depend strongly on VPD but then the multiplier of VPD (b1) takes a larger value for NVP's than for the original grasses so this seems a bit counterintuitive. Could you add a bit more explanation here?

* Section 2.2.3: For the NVP's, when you have negative NPP you induce a biomass loss function. But presumably the negative NPP itself should also lead to a biomass loss. I am interested to know how this works - are these are somehow linked or are they two separate loss terms?

* Section 2.2.3 and Figure 1. Why did you reduce the turnover again after a certain amount of time? (ie why does the line on figure 1 decrease again after it reaches its maximum?) It would be helpful to provide some evidence from the literature or some

more scientific justification here.

* P6 A few issues around equation 4 (which is labelled as 3 by the way!). Underneath the equation you wrote "b is the daily leaf biomass" but this is in units of $gCm^{-2}$, which doesn't have any units of time, so it isn't 'daily'? Do you mean the value gets updated daily? I suggest removing the word 'daily' here. However, there should be some units of time in the turnover rate and I think these might actually be in lcoef, which you have given as no units, I think this should maybe have units of $day^{-1}$ or similar? Difficult for me to tell from the information here but please check it. Another point about this equation, why does it only apply when LAI>LAI_max instead of LAI>LAI_lim? Using LAI_max means it will jump from zero when you reach LAI_max, whereas if you start turning over when it reaches LAI_lim, it will increase smoothly from zero. Maybe this was a typo, but if not, can you explain why you do it? Thanks!

* P8 Equation (10). This is quite a complicated equation and it would be really useful to see what the moisture function actually looks like. I suggest you add a plot of this. I looked in the paper that you referred to but it was not easy to immediately see it, and the moisture function for respiration is important so would be great to include the plot here.

* P9 line 4/5 says that albedo and roughness were set the same as C3 grasses. I guess for NVP's the roughness could be quite different from grass? Could you add a comment on possible differences? Either here or in the discussion.

* P10 Equation 11a) The text says it's a logarithmic function, but this does not seem to be the case? Equation 11b) Bottom line of fraction should have $D^{\gamma}$ not $D^2$ Given these equations, I am not sure it makes sense to fix the crown area but still vary the biomass and height. This means that the allometric relations don't hold (for the case without dynamic vegetation), because the allometric relations are basically the relationship between height and area (or diameter- but these are related), yet you are varying the height and not the area. Could you comment on this? Are you assuming

that the number of individuals changes in order to keep the crown area fixed? If so, please make that clearer in the text.

* P10 Section 2.3.2 In the introduction you said that shrubs accumulate more snow in winter than trees (p3 line 13), but in this section you seem to treat them both together. What is the reason for this?

* P10 equation (13) I can guess what you are doing here - assuming that with very few shrubs they'll be spread out so they won't accumulate much snow, and with a lot of shrubs of course the snow will be the same as the grid box mean because they are covering the whole grid box. But what is the justification for peaking in the middle? Maybe with just a few shrubs they would still accumulate snow? Did you get this function from somewhere or did you come up with it yourself? Could you either (in the first case) add a reference or (in the second case) give a bit more explanation of the physical reasoning?

* P11 Equation (15) I am not sure I agree about the form of this. Because you are integrating, the mortality rate (as a fraction of biomass) depends on the height of the shrub. Imagine your temperature is just constant with z, then the mortality rate will be proportional to (H-Hmin) and thus higher for a taller shrub - despite both being at the same temperature. Is this something you wanted to include in the model? If so, you should discuss it. If not, I would suggest you instead divide the RHS of the equation (15) for Mce by (H-Hmin).

* P12 first paragraph: I don't quite understand what f_v_max is. Do you prescribe a certain fraction of the grid cell to be occupied by a PFT but then it doesn't necessarily occupy that whole fraction? Please explain this term a bit more.

* P12 equation (17) - you do the weighted average in terms of 'log's, I assume this is standard procedure from somewhere but I haven't see it before. Please add a reference.

* P13 L21-24 not sure what you mean by these things: - "survival or estabilishment limits" - limits in terms of what? Temperature? - "a cumulated degree-day threshold for the development" - maybe here you mean "..for the development of leaves"?

* P14 line 1, talks about methods for wetlands, but surely not all of your sites are wetlands?

* P14 line 32/33, it seems odd that the Arctic grasses are assigned to cold climates but then they all end up in the South! Have you checked this?

* P15 line 7/8 What was the justification for these new distributions, especially with the grass fraction. Why did you include grass but not include any shrubs? Also a bit concerning that your percentages don't add up to 100%. What is the rest?

* P16 : last sentence in section 2.6.1 talks about simulations and spin-up with no context (eg forcing data, soil characteristics?). I assume that the same simulation protocol as described in 2.6.2 is used for these simulations, and you extract the closest grid cells? But then the start of the simulation that it refers to at the end of Section 2.6.1 is not the same as described in Section 2.6.2. You need to more clearly explain what simulations are done/used for the parameter optimization.

* Section 3.1 - the first 3 lines here are more like methods than results. Can you make this an extra (final) section in the methods perhaps?

* P17 line 23 How do you know the water stress in the model is too large? Could you show some evidence for this, or that it was seen in previous studies with ORCHIDEE?

* P20 line 3/4 "too low LAI seems to be simulated in western Siberia" This looks more like the middle of Siberia to me?

* P22 line 14 "plant resistance to water stress" - I thought you added something that made the NVP's recover more slowly from drought, and lose biomass, rather than resist the drought. Sorry if I missed the point here - do the other types of plants instead die in those circumstances? If so, could you clarify this?

* P22 line 32 "especially for NVP's" - Not sure about this. Aren't NVP's less nitrogen limited than other plants?

* P22 at the bottom of the page, you are talking about splitting shrubs into different types. It would be helpful to add in a comment about why it would be useful to do this? (What impact it might have?)

* P23 line 14/15, you are talking about how the seasonal cycle of NVP productivity differs from the vascular plants in the model, but there is no comment about whether these differences are realistic. You also mentioned earlier in the paper about 'representing the observed temporal dynamics of lichen and bryophyte biomass', but no reference to actual observations. It would be helpful to refer to some studies to discuss whether the behaviour of the model is realistic.

* P23 line 37/38, "the new PFTs are more sensitive to climate change than the original ones" - the plots do not seem to fully support this. The fractional changes are maybe larger with the new PFT's, but the 'old' PFT that you show on the plot (boreal broad-leaved trees) seems to have the largest absolute change and so potentially the biggest impact on the carbon cycle. I recommend modifying this discussion to account for this.

* P25, Acknowledgements - I suggest you add more details of the projects, not just the acronyms i.e. full names and project numbers.

* P38 Table 5. I think it is interesting that one of the calibrated parameters (b) was calibrated to zero. This appears to remove the acclimation behaviour from the photo-synthesis model. Could you add a comment about this in the text? Do you think it's because the air temperature never gets very warm so acclimation isn't necessary?

Technical comments (In general the writing is good but I picked up some grammar/typos on the way through so will list these here.)

* P1 Line 24, "transpiration (+33%)" -> "transpiration (-33%)"

* P2 Line 23/24, "is relatively simple and discretized on few" -> "has been relatively

simple, with few"

* P2 Line 26 "either trees or grasses PFTs." -> "either trees or grasses."

* P2 line 27 "in the reality" -> "in reality"

* P2 line 36 "interactions part" -> "interactions as part"

* P3 line 4, I'm not sure about how you have referenced the CAVM, you have written "Mapping Team et al.", I wonder if it should just be "Mapping team" (and then the names listed are the members of the mapping team, not additional people?)

* P3 line 7 "does not allow to" -> "does not allow it to"

* P3 line 9 "mosses and lichens and shrubs" -> "mosses, lichens and shrubs"

* P3 line 12 "more resistant for hydric" -> "more resistant to hydric" And "or for nitrogen limitation" -> "or to nitrogen limitation"

* P3 line 15 "to warming whereas trees" -> "to warming, whereas trees"

* P4 line 16 "C3 grasses plants" -> "C3 grasses"

* P6 line 1 "cold temperatures" -> "cold temperature"

* P6 line 33 "(use in ORCHIDEE)" -> "(used in ORCHIDEE)"

* P7 line 13 "when NVP get desiccated." -> "when NVPs get desiccated."

* P7 line 30 "NVPs layer" -> "NVP layer"

* P8 line 24 "to define the control litter" -> "to control litter"

* P9 line 12 "processes as trees." -> "processes to trees."

* P9 line 22 "additional shrubs types" -> "additional shrub types"

* P11 line 2 "dynamically the vegetation distribution" -> "the vegetation distribution dynamically"

* P11 equation 16 Change 'else' to 'otherwise'

* P12 line 6 "there is no woody" -> "there are no woody"

* P12 line 27 "equation described previously" -> "equations described previously"

* P12 line 27 "as well as few" -> "as well as a few"

* P12 line 29 "Cold climates" -> "Cold climate"

* P12 line 34 "themselves function of" -> "themselves functions of"

* P13 line 12 "list of variable" -> "list of variables"

* P13 line 31 "observations located in" -> "observations are located in"

* P16 line 9 "number of iteration" -> "number of iterations"

* P18 line 12 "referred as" -> "referred to as"

* P20 line 25 "occur in early spring" -> "occurs in early spring"

* P20 line 27 "impact the albedo" -> "impacts the albedo"

* P20 line 34 "Contrariwise" -> "Conversely"

* P21 line 14 "5mmd-1" should be "0.5mmd-1" ?

* P21 line 20 "permanent frozen soil" -> "permanently frozen soil"

* P22 line 27 "implies to introduce" -> "implies introducing"

* P22 line 30 "availably" -> "availability"

* P23 line 19 "on the same time" -> "at the same time"

* P24 line 7 "in liason with" -> "in conjunction with"

* P24 line 13 "ecosystem occur" -> "ecosystems occur"

* P24 line 23 "permafrost extension" -> "permafrost extent"

* P24 line 33 "soil water dynamic" -> "soil water dynamics"

* P25 line 5 "and reach around" -> "and reaches around"

* P25 line 11 "reduce locally" -> "locally reduce"

* P25 line 12 "snow dynamic" -> "snow dynamics"

* Table 2 (df) "Maximum number of day for this extra turnover" -> "Maximum number of days ..."

* Table 3 caption "values are choose" -> "values are chosen"

Hope you find this helpful! Best wishes.

---

## Referee Comment (RC2) · Anonymous Referee #2 · 17 May 2017

Druel et al. include a number of new processes and parametrizations into the land surface model ORCHIDEE that are thought to be important in high latitude ecosystems including • parameter optimization of C3 grass, • implementation of a new shrub PFT, and • implementation of a new PFT representing lichens and bryophytes.

Several additional relationships and processes have been also included, such as • shrub-snow interactions, • vertical soil organic matter profile, • moisture dependence of heterotrophic respiration and anoxic conditions, • moss effects on thermal diffusivity.

In general, I fully agree with the importance to advance the LSMs in these respects and I would like to see such important model development published soon. The authors also use a number of site-level observations and a formal parameter calibration procedure for this model development. However, I have some serious concerns about this manuscript which should be addressed prior to publication.

Most importantly, there are too many different topics treated in this single manuscript which then are themselves mostly only superficially addressed and which even may not have any relation to each other (in the model). I strongly suggest to focus the paper on 1-2 research questions and a reduced amount of new processes added. I would agree with a presentation of new shrub, moss and C3 grass parametrizations. After a thorough model evaluation, some model application could be presented e.g. to understand the relation of their carbon balances to each other and to trees as well as their effects on soil temperature. Still, I believe individual papers for shrub and moss functions and effects would be more clear. If all topics should stay within one paper, then substantial additional text and figures/tables are required in order to i) explain the research question and importance of processes using literature, ii) evaluate new (and sometimes old if affected) model functions, iii) present and discuss results with recent literature, and maybe apply the model to address a research question.

Some detailed important issues:

0) It is unclear to me how the authors can neglect the recent publication by Porada et al. (2016) which presents a process-oriented and dynamic representation of bryophytes and lichens in the land surface model JSBACH in introduction and discussion.

1) Mosses have an important function in Boreal forests and the forest ground is usually covered by mosses and lichens. Usually we can expect a NVP cover of more than 50% in Boreal forests and more in tundra (Rapalee et al., 2001; Porada et al., 2016). The approach in this study is to treat NVPs as separate PFT with a separate tile results in minor coverage in most regions. (The color scale in fig 5 is not useful to evaluate the shrub and moss cover, please improve). Hence, there will be a strong bias in moss and lichen effects on the heat balance and biogeochemical ecosystem functions using

such model. That limitation should be discussed in detail.

2) I agree with the authors that the global model can hardly cover small-scale variations in NPP and biomass of shrubs and mosses and lichens. Therefore, I suggest modify Fig 6 such that we see one dot for each climatic zone representing the model and data means but including error bars representing their std. Then one can discuss where the model fails to reproduce natural variance within one climatic zone and natural variance among zones. Fig 7 shows importantly that there is hardly any latitudinal variation in the measurements while the model shows a strong variation. Please, discuss in detail.

3) It seems, model calibration and evaluation at site level has been performed with the same data. If you have too little data to split the dataset into representative parts for calibration and evaluation, then please repeat the site-level model evaluation with a bootstrap method: iteratively remove data for calibration and evaluate respective model results at these sites.

4) I do not agree that LAI is a valid dataset from remote sensing data which is useful for process model evaluation (and if you like to use it please show in the fig ORCH13-GLASS and ORCH16-ORCH13 in order to understand the previous model bias and improvement). Possible maps for a landscape-scale model evaluation: fAPAR (JRC), GPP (Jung et al., 2011 or Beer et al., 2010), evapotranspiration (Jung et al., 2010), biomass (Thurner et al., 2014), and inventory-based NPP and biomass data (IIASA; Beer et al., 2006; Quegan et al., 2011). This is important as the fraction of tiles of all PFTs has been modified. In general, it would also really good to evaluate catchment runoff with freely available data of large Arctic rivers.

5) The reduction in tree cover results in a reduction of transpiration in your grid cell averages. However, interception loss and evaporation should increase with a layer of mosses and lichens. If the water and energy balance is a topic in your paper, then please show results for all components, not only transpiration in Fig 12.

6) In this model version, two modifications affect soil temperature: snow depth and
moss&lichen cover. First of all, the model version should be evaluated in terms of snow depth and soil temperature. For soil temperature, you can use GTN-P borehole data from Romanovsky et al. (2010) and Christiansen et al. (2010) available at PANGAEA, and maps of soil temperature and ALT even from your study region from Beer et al. (2013) at PANGAEA. I expect a cooling effect from mosses (Porada et al. 2016) due to higher insulation in summer, and a warming effect due to higher snow depth in areas of high shrub cover (still unclear to me at landscape scale as shrubs accumulate snow from lateral wind transport, so it is just relocated within the grid cell?). In Fig 13 both effects are combined. Is there a way to separate them? In Fig 13 it seems the model overestimates ALT and that is even higher in ORC16? In Fig 13b it seems all three grid cells show higher ALT (red) while in 13c one profile shows warmer temp (red) and the others show cooler temp? I generally suggest concentrating on soil temperature because ALT estimation from modelled temperature is not reliable.

7) Parameter estimation: Please show a priori and a posteriori parameter distributions in the appendix.

8) Please include a discussion section in which you interpret the results using literature in order to learn something. Parts of your summary section can be used if enhanced by literature. The conclusions and outlook section should be much reduced.

9) Several new methods are described but their importance, evaluation, and application is unclear: • Section 2.2.6: anoxic conditions are not simulated, soil organic matter dynamics are no topic of the paper. Please remove. Or was the intension to evaluate GPP and NEE at eddy covariance sites? • Why is shrub allometry important and why not only assume smaller trees? • Shrub-snow interactions are not evaluated or analyzed. What do we learn from these additional functions? • Effects on albedo: Has been albedo improved when comparing to satellite products?

Minor issues:

Fig 10: not used in results but only in summary and that there also the fig does not

support the sentence.

$CO_2$ conductance in non-vascular plants depends strongly on its moisture and not on stomatal conductance. If that concept is not used here, then please discuss this limitation and related potential biases in detail.

Page 16, line 35: I do not understand.

---

## Referee Comment (RC3) · Anonymous Referee #3 · 19 May 2017

Review of:

Towards a more detailed representation of high-latitude vegetation in the global land surface model ORCHIDEE (ORC-HL-VEGv1.0)

By Arsène Druel et al.

This manuscript describes a revision to the ORCHIDEE land surface model to improve the way in which tundra and subarctic vegetation are simulated by the model. The authors achieve this update by implementing three new plant functional types (PFTs) – these are a boreal shrub type, an arctic graminoid type, and a non-vascular plant type – into the model framework. Implementing new PFTs in ORCHIDEE has two steps,

1) changing process representations where necessary, and 2) defining the set of parameters that characterizes each PFT. The new shrub and grass PFTs needed few changes to process representation to implement, while on the other hand, simulating the non-vascular plant PFT required a different way of dealing with plant water uptake, gross productivity, and mortality. Parameter sets for each of the new PFTs were estimated using a Bayesian estimation process. The authors use the result of the new PFTs, updated process representations, and parameter sets and run the new version of the (ORC16), and compare the result to field-based observations, to satellite remote sensing products, and to the previous version of the model (ORC13) to highlight the effects of the update.

In general, this manuscript is valuable and should be published. It describes a valuable update to ORCHIDEE, which will undoubtedly be used in a number of forthcoming and future studies. The changes to the model lead to improvements in the comparison with observations, and thus represent progress over ORC13. However, the manuscript presentation is not particularly good: the text requires a thorough copyediting to clarify grammar and usage style, some of the figures are too small, and there a few small issues concerning the presentation of units and values which are elaborated below. Aside from these presentation issues, my major concern of this study was the choice of data used to inform the parameter optimization, and the appropriateness of comparing site-level measurements with model simulations performed on a 2-degree grid.

The largest concern I have with the current study is the authors' apparent inability to assemble a larger, more representative dataset of high-latitude plant characteristics with which to parameterize the model. Their Bayesian optimization relies exclusively on the Peregon et al., 2008 biomass and NPP dataset. These data were specifically collected on *wetland* vegetation, while ORCHIDEE, in this paper, is intended to simulate *upland* vegetation. This mismatch between what the data represent and what the model is trying to simulate is a very serious limitation and calls into question the appropriateness and quality of the model parameterization. Use of such a limited and specialized

dataset to parameterize a global model might be acceptable in regions of the world for which there are very few ecological and ecophysiological data, e.g., in parts of the tropics, but for the Arctic, it is practically inexcusable because of enormous amount of field research that has been performed over the last 50 years. Data from iconic arctic research sites such as Toolik Lake in North America, Abisko in Europe, and Zackenberg in Greenland were ignored in development of the testing dataset. Large amounts of data on key characteristics such as aboveground biomass were collected in the entire circumarctic region as part of, e.g., the ITEX experiment. Data from all of these locations outside of west Siberia, while perhaps more difficult to assemble, could have provided valuable information on the status of upland tundra and subarctic vegetation that would have been more appropriate for performing the model parameterization. If the authors prefer to not improve their parameterization using more widespread and representative field data, at very least they should explain and justify their choice for using the wetland dataset of limited spatial extent more clearly in the manuscript.

Specific comments

Page 2, line 3
The last glacial inception began around 126.5-120 ka; correct this error

Page 2 line 28
The model described is called BIOME4; please correct the model name

Page 8 line 16
Anoxic conditions affect the activity of all types of soil microorganisms, not only bacteria, e.g., fungi, archaea, and multi-celled microorganisms. Please be more inclusive instead of using the word "bacteria"

Page 13 line 14-18
Why not make the root profile shape parameter a function of the mean active-layer thickness? The model simulates active layer thickness, and presumably most plants would optimize their rooting profile to be compatible with this value

Page 13 lines 21-23
This sentence is confusing. Please revise for clarity by explaining how this version of ORCHIDEE uses prescribed vegetation cover and therefore survival and establishment limits are not relevant.

Page 13 line 31
Explain why using observational data collected in "boreal wetlands" is appropriate for a parameterizing a global model that simulates predominantly upland systems, indeed, there is no representation of wetlands at all in this version of ORCHIDEE (as far as I could understand).

Page 14 line 14-16
If the model was run on a 2-degree grid, why were the site-level data aggregated only to half-degree? Wouldn't it have made more sense to aggregate the data at the same spatial scale as the model simulations? Also, the choice of dataset (from wetlands) clearly limits the amount of data coming from non-vascular plants, shrubs, and grasses; wouldn't an effort to assemble a more spatially global and upland-representative dataset have helped here?

Page 16 line 4-5
The phrase starting ". . .in CAVM Mapping Team. . ." is awkward and hard to understand. Rephrase.

Page 16 line 20-21
As we know multi-annual and decadal climate cycles exist, e.g., ENSO, and that there was a clear trend on climate during the 1st half of the 20th Century, is it appropriate to select individual years randomly over this period for the model spinup? I realize that many other vegetation modeling protocols prescribe the same thing, but that doesn't mean that it is correct. Using a detrended climate timseries would be a minimum first step towards improving the quality of the model spinup.

Page 17 line 24

If the 2-degree resolution used to run the model presents problems in terms of comparison with observations, why wasn't the model run at finer resolution, or in an "individual point" model with local forcing. This version of ORCHIDEE does not simulate any 2D spatial processes that would be impossible to implement in a point mode.

Page 17 line 32-34
Making an effort to assemble a larger calibration-evaluation dataset would have helped here. If these data really do not exist, this has to be clearly explained in the manuscript.

Page 18 line 1-2
Again, having more, and more widespread observations might have helped here.

Page 18 line 24-26
I would be very helpful for the reader if the meteorological variables were provided in terms of more ecologically relevant units. For example, provide precipitation in terms of annual totals, and temperature in terms of summertime (JJA) or growing season means (instead of annual? – it's not clear what is provided here).

Page 19 line 16-17
Again, what are these temperature anomalies referring to – seasonal, annual, individual months? A +10 anomaly in winter temperature in an place where the mean winter temperature is -40 C may not really be ecological relevant.

Page 20 line 3-4
The phrase with "...too low LAI..." is awkward. Revise.

Page 21 line 1-2
Again, provide ecologically relevant units, e.g., total transpiration per month.

Page 21 line 11-14
Again, adjust units of evapotranspiration, runoff, etc. to monthly, seasonal, or annual sums. Annual is probably best here.

Page 22 line 35

In the boreal regions and Arctic, the shrub vegetation is composed of both evergreen and deciduous (summergreen) broadleaved plants (angiosperms), and evergreen needleleaf plants (gymnosperms). Thus, there are at least three types of shrubs.

Figure 5
The maps should be reproduced in a larger size

Figure 6
The plots should be reproduced in larger size, or at least the points should be plotted a bit larger. It is hard to see some of the points, especially the cyan colored dots

Figure 11
The maps should be reproduced in a larger size

Figure 12
The maps should be reproduced in a larger size
* * *

---

## Author Comment (AC2) · 16 Sep 2017

**Response to Referee #3**

*We thank R3 for this helpful review. Enclosed please find a detailed explanation of the revisions we made based on R3's comments. For your convenience, comments are in bold and our response is in italic. Revisions we made in the manuscript are presented in italic with grey background.*

**This manuscript describes a revision to the ORCHIDEE land surface model to improve the way in which tundra and subarctic vegetation are simulated by the model. The authors achieve this update by implementing three new plant functional types (PFTs) – these are a boreal shrub type, an arctic graminoid type, and a non-vascular plant type – into the model framework. Implementing new PFTs in ORCHIDEE has two steps, 1) changing process representations where necessary, and 2) defining the set of parameters that characterizes each PFT. The new shrub and grass PFTs needed few changes to process representation to implement, while on the other hand, simulating the non-vascular plant PFT required a different way of dealing with plant water uptake, gross productivity, and mortality. Parameter sets for each of the new PFTs were estimated using a Bayesian estimation process. The authors use the result of the new PFTs, updated process representations, and parameter sets and run the new version of the (ORC16), and compare the result to field-based observations, to satellite remote sensing products, and to the previous version of the model (ORC13) to highlight the effects of the update.**

**In general, this manuscript is valuable and should be published. It describes a valuable update to ORCHIDEE, which will undoubtedly be used in a number of forthcoming and future studies. The changes to the model lead to improvements in the comparison with observations, and thus represent progress over ORC13. However, the manuscript presentation is not particularly good: the text requires a thorough copyediting to clarify grammar and usage style, some of the figures are too small, and there a few small issues concerning the presentation of units and values which are elaborated below. Aside from these presentation issues, my major concern of this study was the choice of data used to inform the parameter optimization, and the appropriateness of comparing site-level measurements with model simulations performed on a 2-degree grid.**

*The entire manuscript was re-read by a native English speaker. Possible further improvements may be done upon request for a next stage of revision*

*or upon acceptance. However the grammar and usage style changes are not reported in this response. Furthermore, the size of all figures has been increased. Concerning your other comments, please find some answers below.*

**The largest concern I have with the current study is the authors' apparent inability to assemble a larger, more representative dataset of high-latitude plant characteristics with which to parameterize the model. Their Bayesian optimization relies exclusively on the Peregon et al., 2008 biomass and NPP dataset. These data were specifically collected on *wetland* vegetation, while ORCHIDEE, in this paper, is intended to simulate *upland* vegetation. This mismatch between what the data represent and what the model is trying to simulate is a very serious limitation and calls into question the appropriate- ness and quality of the model parameterization. Use of such a limited and specialized dataset to parameterize a global model might be acceptable in regions of the world for which there are very few ecological and ecophysiological data, e.g., in parts of the tropics, but for the Arctic, it is practically inexcusable because of enormous amount of field research that has been performed over the last 50 years. Data from iconic arctic research sites such as Toolik Lake in North America, Abisko in Europe, and Zacken- berg in Greenland were ignored in development of the testing dataset. Large amounts of data on key characteristics such as aboveground biomass were collected in the en- tire circumarctic region as part of, e.g., the ITEX experiment. Data from all of these locations outside of west Siberia, while perhaps more difficult to assemble, could have provided valuable information on the status of upland tundra and subarctic vegetation that would have been more appropriate for performing the model parameterization. If the authors prefer to not improve their parameterization using more widespread and representative field data, at very least they should explain and justify their choice for using the wetland dataset of limited spatial extent more clearly in the manuscript.**

*We are aware of your concern about the spatial representativeness of the dataset used for the Bayesian optimisation. However part of our choice is justified by specific needs for the calibration and by the accessibility to the data. We needed total living biomass and productivity at different sites, with multi-annual observations, and for the three new PFTs. The published and unpublished data provided by Peregon et al. satisfied these criteria, while we did not find easily other data sets satisfying all criteria. We agree that there is a large amount of recent campaigns in the Artic with numerous in situ measurements especially at specific highly instrumented sites; however these data are not assembled into a freely available and comprehensive database. Note also that the western Siberian data are acquired mainly on lowlands but*

not exclusively on very humid sites. As you suggested, in this case it is important to clarify our approach and we have thus added in the text: p.14 l.36-38: "Note finally that using a single dataset in Western Siberia (mainly lowlands) for the model calibration may introduce some biases, which will have to be evaluated."

However, to account for your very relevant suggestion, we have searched for additional data for the model evaluation, especially from the sites you recommended. We did not find any complete data set, in the mass of published literature, which could be used easily for the optimization step along with the Siberian data. Nevertheless, we now use two North-South Arctic transects (with biomass data in lowlands and uplands): one in Eurasia (Walker et al, 2011a) and one in North America (Walker et al, 2011b; and previous reports since 2007). While these data are not sufficient for the optimisation (the productivity is missing), we propose to use them to evaluate the model. We do not claim that a larger set of data could not have been gathered but given the focus of the paper, i.e. on the new process description, we believe the two sets of data that are now used (from Western Russian and from two transects) are sufficient. We added a new figure (Fig. 9) for the model evaluation with associated comments reported below. Note finally that we discuss in the text the potential shortcomings due to the use of mainly lowland data for the calibration of a global model.

P.17 l.14-19: "We further compare the simulated biomasses with two other Arctic transects. The first one is the North America Arctic Transect (NAAT). It is situated in a continental area, and includes eight field locations (70°N 149°W to 79°N 100°W) sampled from 2002 to 2006 (Walker et al., 2011b) chosen as representative of zonal conditions. The second, located in a marine-influenced area, is the Eurasian Arctic Transect (EAT). It includes six field locations (58 to 73°N, between 67 to 81°E) sampled from 2007 to 2010 (Walker et al., 2008, 2009a, 2009b, 2011a)."

P.19 l.1-15: "Carbon stock with two Arctic transect

To evaluate the modelled biomass in other Arctic sites (not used in the calibration step), including uplands and lowlands, Fig. 9 shows scatter plots of observed and simulated biomass along two transects: the NAAT (North America) and the EAT (Eurasia) Artic Transect. The NVPs and shrub biomasses are relatively well reproduced by the model (i.e. within the error bars). For both PFTs, the standard deviation of the observations includes the 1:1 line, but the observed biomasses are on average higher than the simulated biomasses. Simulated shrub biomasses are biased low for the NAAT transect but not for the EAT transect.

*In contrast, the mean value of observed biomass for boreal C3 grasses (Fig. 9.c) is low compared to the simulated biomasses for both cases. For half of the sites the simulated low biomass is in accordance with the observations, but for the other half the values are much larger (> 300 gC.m² whereas the observation do not exceed 54 gC.m²). Despite the optimization with observations from western Siberia (Fig. 7; leading to a decrease of biomass compared to temperate C3 grasses) there is likely an overestimation of the biomass for boreal C3 grasses, probably associated with an overestimated productivity."*

*Walker et al, 2011a: Vegetation of zonal patterned-ground ecosystemsalong the North America Arctic bioclimate gradient. Applied Vegetation Science 14, 440–463. Doi: 10.1111/j.1654-109X.2011.01149.x*

*Walker et al, 2011. 2010 Expedition to Krenkel Station, Hayes Island, Franz Josef Land, Russia, Data Report, Alaska Geobotany Center, Institute of Arctic Biology, University of Alaska Fairbanks, Fairbanks, AK. 63 pp.*

**Specific comments**

**Page 2, line 3**

**The last glacial inception began around 126.5-120 ka; correct this error**

*Done*

**Page 2 line 28**

**The model described is called BIOME4; please correct the model name**

*Done*

**Page 8 line 16**

**Anoxic conditions affect the activity of all types of soil microorganisms, not only bacteria, e.g., fungi, archaea, and multi-celled microorganisms. Please be more inclusive instead of using the word "bacteria"**

*We thank the reviewer for this relevant comment. We changed "bacteria" for "soil microorganism" (p9. l.1)*

**Page 13 line 14-18**

**Why not make the root profile shape parameter a function of the mean active-layer thickness? The model simulates active layer thickness, and presumably most plants would optimize their rooting profile to be compatible with this value**

*We agree that this is an important suggestion. Using a dynamical root profile*

*water table or the plant growth status. However, to keep model consistency, it should be applied to all PFTs of the model, and not only the PFTs developed in this study. Given the requested work especially for the calibration issues, we chose not to change the general equation of the current version of ORCHIDEE. Note that this is currently under investigation for all PFTs.*

**Page 13 lines 21-23**

**This sentence is confusing. Please revise for clarity by explaining how this version of ORCHIDEE uses prescribed vegetation cover and therefore survival and establishment limits are not relevant.**

*We changed the sentence to: "Note that we did not add any bioclimatic limits, such as i) survival or establishment temperature thresholds as proposed by Bonan et al., (2003) and Oleson et al. (2013) or ii) a cumulated degree-day threshold (above the zero degree criteria) for the plant growth (Miller and Smith, 2012). In this study we use ORCHIDEE without the dynamic vegetation module, but with a prescribed vegetation cover preventing vegetation development in unfavourable areas" (p.14 l.5-9).*

**Page 13 line 31**

**Explain why using observational data collected in "boreal wetlands" is appropriate for a parameterizing a global model that simulates predominantly upland systems, indeed, there is no representation of wetlands at all in this version of ORCHIDEE (as far as I could understand).**

*We have already partially answered this comment above. The first reason is that it was the most appropriate dataset that was available to us, even though it concerns mainly lowlands. Secondly, although it is considered as lowland on average, such data set comprises some sites that are not so-called wetlands. Finally, although we have kept this data set for the model calibration, in order to evaluate results at a global scale, we now use an additional set of observations for the model evaluation. These new data include both upland and wetland observations (Fig. 9 and associated comment, p.17 l.14-19, p.19 and l.1-15).*

**Page 14 line 14-16**

**If the model was run on a 2-degree grid, why were the site-level data aggregated only to half-degree? Wouldn't it have made more sense to aggregate the data at the same spatial scale as the model simulations? Also, the choice of dataset (from wetlands) clearly limits the amount of data coming from non-vascular plants, shrubs, and grasses; wouldn't an effort to assemble a more spatially global and upland-representative dataset have helped here?**

*For the first part of your comment, indeed this point was not clear enough. In fact, the optimization is also done at 0.5° resolution. We have now added a new sentence to clarify this in the Section about the optimization (2.6.1): "The simulation for the optimisation was done with CRU-NCEP meteorological forcing (Wei et al., 2014; Viovy, 2015), at 0.5° resolution" (p. 16, l.36-37 and p.17 l.1).*

*The second part of your comment was already answered above.*

**Page 15 line 4-5**

**The phrase starting ". . .in CAVM Mapping Team. . ." is awkward and hard to understand. Rephrase.**

*We rephrased as follows: "In the map from Loveland et al. (2000), we noticed that the tundra biome corresponds to the "sparse vegetation" or to the "lichens and mosses" LCCs distribution. In CAVM Mapping Team (2003), the tundra biome is described as containing ~30 to 60% NVPs." (p.15, l.39-31)*

**Page 16 line 20-21**

**As we know multi-annual and decadal climate cycles exist, e.g., ENSO, and that there was a clear trend on climate during the 1st half of the 20th Century, is it appropriate to select individual years randomly over this period for the model spinup? I realize that many other vegetation modeling protocols prescribe the same thing, but that doesn't mean that it is correct. Using a detrended climate timeseries would be a minimum first step towards improving the quality of the model spinup.**

*Thank you for this remark. Indeed this is probably a better solution. However, for this study, this would lead to re-running all simulations, which was not possible at this stage. Moreover, the impact on above-ground boreal vegetation after a century of stable climate would probably be minor.*

**Page 17 line 24**

**If the 2-degree resolution used to run the model presents problems in terms of comparison with observations, why wasn't the model run at finer resolution, or in an "individual point" model with local forcing. This version of ORCHIDEE does not simulate any 2D spatial processes that would be impossible to implement in a point mode.**

*The aim of this study was to improve boreal representation on a global scale. However, at such scale, fine-resolution (e.g. 0.5°, used for the optimization) simulations would be too computationally demanding. Moreover, local (point) meteorological forcing data, including precipitation, temperature, downward longwave and shortwave radiation, relative humidity and wind, were not*

*climate forcing based on a merge of climate reanalysis and in situ observations. Else we agree that if the local forcing data would have been available, we should have used them.*

**Page 17 line 32-34**

**Making an effort to assemble a larger calibration-evaluation dataset would have helped here. If these data really do not exist, this has to be clearly explained in the manuscript.**

*This comment was already answered above. Data we found for other sites are not complete enough to be used for the calibration. However, they were used to improve the evaluation of our results (Fig. 9 and associated comment, p.17 l.14-19, p.19 and l.1-15).*

**Page 18 line 1-2**

**Again, having more, and more widespread observations might have helped here.**

*Same as above*

**Page 18 line 24-26**

**I would be very helpful for the reader if the meteorological variables were provided in terms of more ecologically relevant units. For example, provide precipitation in terms of annual totals, and temperature in terms of summertime (JJA) or growing season means (instead of annual? – it's not clear what is provided here).**

*We have changed most units to more ecologically relevant ones (in "$mm.y^{-1}.m^{-2}$" in p.21 l.21,29-31,34, p.50 l.12-13, Figs. 12 and S5). Moreover, we have now indicated more clearly on which period temperature is considered ("growing season (AMJ) mean air temperature" p.50 l.13) and we have updated the values (p.50 l.16)*

**Page 19 line 16-17**

**Again, what are these temperature anomalies referring to – seasonal, annual, individual months? A +10 anomaly in winter temperature in an place where the mean winter temperature is -40 C may not really be ecological relevant.**

*It was annual temperature anomalies, but which are present all along the year (winter, growing season or summer). In order to be clearer and consistent with precedent changes, we changed by the growing season: p.51 l.11-12: "growing season temperatures" and "+6°C and + 10°C compared to America and Asia respectively".*

**Page 20 line 3-4**

**The phrase with "...too low LAI..." is awkward. Revise.**

*We rephrased as follows: "However, the model underestimates LAI in the central-west of Siberia" (p.19 l.21).*

**Page 21 line 1-2**

**Again, provide ecologically relevant units, e.g., total transpiration per month.**

**Page 21 line 11-14**

**Again, adjust units of evapotranspiration, runoff, etc. to monthly, seasonal, or annual sums. Annual is probably best here.**

*We change all values by day (in $mm.d^{-1}.m^{-2}$) by values in "$mm.y^{-1}.m^{-2}$" (p.21 l.21,29-31,34, p.50 l.12-13), including the Fig. 12 and S5.*

**Page 22 line 35**

**In the boreal regions and Arctic, the shrub vegetation is composed of both evergreen and deciduous (summergreen) broadleaved plants (angiosperms), and evergreen needleleaf plants (gymnosperms). Thus, there are at least three types of shrubs.**

*Indeed, we forgot one type here. This sentence was changed to: "Concerning shrubs, we selected a boreal broad-leaved deciduous phenology, although in reality there is a mix of deciduous and evergreen broadleaf shrubs and evergreen needled-leaf shrubs." (p. 23, l. 26-27)*

**Figure 5**

**The maps should be reproduced in a larger size**

**Figure 6**

**The plots should be reproduced in larger size, or at least the points should be plotted a bit larger. It is hard to see some of the points, especially the cyan colored dots**

**Figure 11**

**The maps should be reproduced in a larger size**

**Figure 12**

**The maps should be reproduced in a larger size**

*All of the designed figures have been enlarged. In particular, the figure 6 has been improved to be more readable.*

---

## Author Comment (AC5) · 16 Sep 2017

**Response to Referee #2**

*We thank R2 for this detailed review. Enclosed please find a detailed explanation of the revisions we made based on R2's comments. For your convenience, comments are in bold and our response is in italic. Revisions we made in the manuscript are presented in italic with grey background.*

**Druel et al. include a number of new processes and parametrizations into the land surface model ORCHIDEE that are thought to be important in high latitude ecosystems including * parameter optimization of C3 grass, * implementation of a new shrub PFT, and * implementation of a new PFT representing lichens and bryophytes.**

**Several additional relationships and processes have been also included, such as * shrub-snow interactions, * vertical soil organic matter profile, * moisture dependence of heterotrophic respiration and anoxic conditions, * moss effects on thermal diffusivity.**

**In general, I fully agree with the importance to advance the LSMs in these respects and I would like to see such important model development published soon. The auhors also use a number of site-level observations and a formal parameter calibration procedure for this model development. However, I have some serious concerns about this manuscript which should be addressed prior to publication.**

**Most importantly, there are too many different topics treated in this single manuscript which then are themselves mostly only superficially addressed and which even may not have any relation to each other (in the model). I strongly suggest to focus the paper on 1-2 research questions and a reduced amount of new processes added. I would agree with a presentation of new shrub, moss and C3 grass parametrizations. After a thorough model evaluation, some model application could be presented e.g. to understand the relation of their carbon balances to each other and to trees as well as their effects on soil temperature. Still, I believe individual papers for shrub and moss functions and effects would be more clear. If all topics should stay within one paper, then substantial additional text and figures/tables are required in order to i) explain the research question and importance of processes using literature, ii) evaluate new (and**

**sometimes old if affected) model functions, iii) present and discuss results with recent literature, and maybe apply the model to address a research question.**

*The organization of the manuscript was an important step ahead of actually writing this article. And as you suggest, we had to decide between isolating in different articles the different boreal vegetation types (PFTs), or writing an article about the global improvement of boreal vegetation (including all PFTs). We chose this second option in light of its submission to GMD, to focus the article on the model implementation at a global scale and not on a model application with in-depth investigation of a scientific question. It must enable users or developers of other LSMs to understand our developments, compare or integrate processes in order to improve global vegetation modelling.*

*In this context, it seems to us that splitting into different articles may reduce the interest of the study, especially in view of the fact that an overall and comprehensive evaluation (with global data) of the implementation would be difficult to split. Similarly, limiting the number of processes described could preclude the global consideration of the new boreal PFTs, and importantly, prevent reproducibility of our developments – being a venue for comprehensive descriptions of new model developments is an important goal of this journal. Finally, to reduce the size and complexity of this article, we chose to keep the application of implementations you suggest, such as the vegetation dynamics, the impact on soil carbon stocks or climate changes, for later articles.*

*However, as you suggested, we have added substantial additional content, especially in the introduction to highlight the research question and appropriate references to the existing literature (p.1 l.31-33, p.2 l.7-11,12,15,20-21,23,26-35,38-40 and p.3 l.6-7,33-37), in the results to provide evaluation on other sites (Fig. 9, p.16-17 l. 38 and l.1-4, and p.18-19 l.32-39 and l.1-6), and in the discussion (from p.22 l.17) to compare our results with more recent studies (in particular Porada et al., 2016). Moreover, in order to clarify and reduce the size of the article, we decided to move the results split by continent (ex figs. 8 to 10 and associated texts) into the supplementary material (Figs. S1 to S3), and to substitute them by Artic-wide averages (Fig. 11).*

*Overall the paper should be considered primarily as a model description with a main focus on non vascular plants and shrubs, while the improvement for C3 grasses reduces to parameter optimization. The evaluation of the new developments at local to continental scales should thus only be considered as a first step to evaluate the potential of a more realistic description of boreal vegetation in a global model and not as an exhaustive evaluation/validation*

*of the carbon, water and energy balance of these ecosystems. Such exhaustive evaluation is not compatible with an in-depth model description in a single paper and is thus left for a subsequent study. However, we have tried to better justify in the paper our choices for the selected evaluation diagnostics (and not all available observations).*

**Some detailed important issues:**

**0) It is unclear to me how the authors can neglect the recent publication by Porada et al. (2016) which presents a process-oriented and dynamic representation of bryophytes and lichens in the land surface model JSBACH in introduction and discussion.**

*Indeed, Porada el al. (2016) is one of the first descriptions of non vascular plants in an ESM, with a process-based implementation. We missed the paper as it only came out after we had already completed our first draft. We thus added this reference in the introduction when describing the current state of boreal PFT in ESM model p.2 l.31-32: "a first description of lichen and bryophytes was implemented in the JSBACH model (Porada et al., 2013), improve recently with a process-based implementation (Porada et al., 2016)". We also compare our results with those of this latter article (in the discussion and conclusion sections) in order to put into perspective our findings.*

**1) Mosses have an important function in Boreal forests and the forest ground is usually covered by mosses and lichens. Usually we can expect a NVP cover of more than 50% in Boreal forests and more in tundra (Rapalee et al., 2001; Porada et al., 2016). The approach in this study is to treat NVPs as separate PFT with a separate tile results in minor coverage in most regions. (The color scale in fig 5 is not useful to evaluate the shrub and moss cover, please improve). Hence, there will be a strong bias in moss and lichen effects on the heat balance and biogeochemical ecosystem functions using such model. That limitation should be discussed in detail.**

*In the version of ORCHIDEE used in this article there is no possibility to take into account and model explicitly the understory vegetation cover (the sum of of all PFTs fraction ≤1). We agree that this poses a severe limitation to fully assess the impact of shrubs and NVPs on ecosystem functioning, and more particularly in boreal landscapes. However, we chose to make a first step with the current model structure, treating NVPs and shrubs as separate PFTs like for the 13 standard PFTS. We should notice that in boreal landscapes the forest cover is relatively sparse with significant gaps, by comparison to temperate or tropical forest cover, thus allowing light to reach the ground*

*more easily. As a first approximation we can thus estimate that NVPs are only partially controlled by the surrounding trees and that the biotic interactions with the other strata are limited.*

*Additionally, treating explicitly the understory vegetation, with a process-based approach, is more complicated as it requires a treatment of the radiation transfer within the canopy that accounts for forest gaps distribution and for the intra-canopy climate. Indeed air humidity and temperature are significantly different above the forest canopy than near the ground. Naudts et al. (2016) made a first crucial step in that direction with the addition in ORCHIDEE-CAN of a 2 streams radiative transfer scheme including a "gap" model and Ryder et al. (2016) further added a multi-layer canopy scheme (for energy, water and carbon fluxes) accounting for intra-canopy climate gradients. Our paper should thus be considered as a first step, describing the main biogeochemical features of NVPs and shrubs (as standalone PFTs), before a more complete and comprehensive integration is made within a vertically discretized canopy model (i.e. the ORCHIDEE-CAN version). We thus decided that the available model structure (at the time of the study) was not sufficient to treat explicitly understory NVPs/shrubs.*

*In this context we agree that the original land cover maps derived from satellite observations largely underestimate the fractional cover of NVPs and shrubs. However, we made an attempt using existing boreal land cover maps to partly correct for this bias. Note also that Peckham et al. (2009) showed that mosses represent a large cover fraction of burned areas, with thus potentially significant year-to-year variations of NVP cover at regional scale. Overall, it was difficult to increase more substantially the NVP/shrub fractional cover without having unrealistically low tree cover. Our study thus represents a lower estimate of the potential impact of NVPs and shrubs on boreal ecosystem functioning.*

*Note finally that the colour scale of Fig. 5 has been improved.*

*Peckham, S. D., Ahl, D. E. & Gower, S. T. Bryophyte cover estimation in a boreal black spruce forest using airborne lidar and multispectral sensors. Remote Sens. Environ. 113, 1127–1132 (2009).*

*Ryder, J., Polcher, J., Peylin, P., Ottlé, C., Chen, Y., Gorsel, E. V., ... & Valade, A. (2016). A multi-layer land surface energy budget model for implicit coupling with global atmospheric simulations. Geoscientific Model Development, 9(1), 223-245.*

Naudts, K., Ryder, J., McGrath, M. J., Otto, J., Chen, Y., Valade, A., ... & Ghattas, J. (2015). A vertically discretised canopy description for ORCHIDEE (SVN r2290) and the modifications to the energy, water and carbon fluxes. *Geoscientific Model Development*, 8, 2035-2065.

**2) I agree with the authors that the global model can hardly cover small-scale variations in NPP and biomass of shrubs and mosses and lichens. Therefore, I suggest modify Fig 6 such that we see one dot for each climatic zone representing the model and data means but including error bars representing their std. Then one can discuss where the model fails to reproduce natural variance within one climatic zone and natural variance among zones. Fig 7 shows importantly that there is hardly any latitudinal variation in the measurements while the model shows a strong variation. Please, discuss in detail.**

*We agree with the reviewer that Figure 6 would benefit from grouping the individual measurements within restricted climatic/geographic zones. We have thus followed this advice and grouped them according to the six subzones.*

*Indeed there is a strong latitudinal variation in the model simulations. However, it seems to us that the latitudinal variation in the measurements is as strong, considering the important variation in the mean as well as in the standard deviation. We therefore regret that we do not understand this comment.*

**3) It seems, model calibration and evaluation at site level has been performed with the same data. If you have too little data to split the dataset into representative parts for calibration and evaluation, then please repeat the site-level model evaluation with a bootstrap method: iteratively remove data for calibration and evaluate respective model results at these sites.**

*We agree with the reviewer that optimally we should always split the dataset into a calibration and evaluation parts. However in our case several constraints arose from i) the relatively small size of the initial dataset for such split and ii) the large computing time necessary for the model calibration which complicates any bootstrap approach (i.e. the calibration took several weeks with the Genetic Algorithm that is used). Given these constraints we searched for additional datasets to fulfil several requests from the different reviewers. We thus now apply the following strategy:*
*1) we keep the original Western Siberia dataset to perform the optimization.*
*2) we use the observations from two new transects in North America and Eurasia (with appropriate biomass data) to perform the model evaluation.*

*We added a new figure (Fig. 9) for the model evaluation with associated comments reported below. Note finally that we discuss in the text the*

*potential shortcomings due to the use of mainly lowland data for the calibration of a global model.*

*P.17 l.14-19: "We further compare the simulated biomasses with two other Arctic transects. The first one is the North America Arctic Transect (NAAT). It is situated in a continental area, and includes eight field locations (70°N 149°W to 79°N 100°W) sampled from 2002 to 2006 (Walker et al., 2011b) chosen as representative of zonal conditions. The second, located in a marine-influenced area, is the Eurasian Arctic Transect (EAT). It includes six field locations (58 to 73°N, between 67 to 81°E) sampled from 2007 to 2010 (Walker et al., 2008, 2009a, 2009b, 2011a)."*

*P.19 l.1-15: "Carbon stock with two Arctic transect*

*To evaluate the modelled biomass in other Arctic sites (not used in the calibration step), including uplands and lowlands, Fig. 9 shows scatter plots of observed and simulated biomass along two transects: the NAAT (North America) and the EAT (Eurasia) Artic Transect. The NVPs and shrub biomasses are relatively well reproduced by the model (i.e. within the error bars). For both PFTs, the standard deviation of the observations includes the 1:1 line, but the observed biomasses are on average higher than the simulated biomasses. Simulated shrub biomasses are biased low for the NAAT transect but not for the EAT transect.*

*In contrast, the mean value of observed biomass for boreal C3 grasses (Fig. 9.c) is low compared to the simulated biomasses for both cases. For half of the sites the simulated low biomass is in accordance with the observations, but for the other half the values are much larger (> 300 gC.m$^2$ whereas the observation do not exceed 54 gC.m$^2$). Despite the optimization with observations from western Siberia (Fig. 7; leading to a decrease of biomass compared to temperate C3 grasses) there is likely an overestimation of the biomass for boreal C3 grasses, probably associated with an overestimated productivity."*

*Walker et al, 2011a: Vegetation of zonal patterned-ground ecosystemsalong the North America Arctic bioclimate gradient. Applied Vegetation Science 14, 440–463. Doi: 10.1111/j.1654-109X.2011.01149.x*

*Walker et al, 2011. 2010 Expedition to Krenkel Station, Hayes Island, Franz Josef Land, Russia, Data Report, Alaska Geobotany Center, Institute of Arctic Biology, University of Alaska Fairbanks, Fairbanks, AK. 63 pp.*

**4) I do not agree that LAI is a valid dataset from remote sensing data which is useful for process model evaluation (and if you like to use it please show in the fig ORCH13-GLASS and ORCH16-ORCH13 in order to understand the previous model bias and improvement).**

**Possible maps for a landscape-scale model evaluation: fAPAR (JRC), GPP (Jung et al., 2011 or Beer et al., 2010), evapotranspiration (Jung et al., 2010), biomass (Thurner et al., 2014), and inventory-based NPP and biomass data (IIASA; Beer et al., 2006; Quegan et al., 2011). This is important as the fraction of tiles of all PFTs has been modified. In general, it would also really good to evaluate catchment runoff with freely available data of large Arctic rivers.**

*As mentioned above, the primary objective of the paper is not to provide a complete and comprehensive evaluation of the model with all potential large-scale datasets, but to provide a complete description of the new PFTs (equations and parameters) including only a first step evaluation.*

*Additionally, the validation of the results by world-scale data is not straightforward and potentially critical. The main problem in proposed global products in that they do not include PFTs (or vegetation) distinctions. Moreover, the biomass, NPP and evapotranspiration are more driven by trees or fire distribution than by the influence of the new PFTs. Comparing these maps with the new vegetation cover could add potentially other sources of bias and thus only little additional information. Moreover, the majority of these data is also derived from satellite observations, with comparable biases to those associated to LAI. The fAPAR product, although less sensitive to saturation issues, also comes with its own issues when comparing to current model outputs. The evapotranspiration product from Jung et al. (2010) may suffer from the small set of eddy-covariance measurements available in the boreal zones.*

*For the catchment runoff, we have done a summary of the river discharge on the ten main Arctic watersheds (http://www.r-arcticnet.sr.unh.edu/v4.0/main.html) to compare with the runoff + drainage simulated on the same area and the same period, p.21 l.19-23: "Compared to observations (main Artic watershed available at http://www.r-arcticnet.sr.unh.edu/v4.0/main.html), the river discharge simulated indicates a general underestimation in the northern high latitudes, linked to an overestimation of evaporation and sublimation (Gouttevin et al., 2012). Thus, this underestimation with ORC16 is smaller than with ORC13."*

*Although not ideal, we thus kept the LAI as a first step evaluation. Following the suggestion of the reviewer, we added a map (and a transect) of ORC16-ORC13 in Fig. 11 (the map ODRC13-GLASS was already showed. That shows a significant difference between ORC16 and ORC13, and so the improvement with ORC16: p19. L.21-22: "This improvement with ORC16 is directly due to significant lower LAI values in these regions (north of 55°N) compared to ORC13".*

**5) The reduction in tree cover results in a reduction of transpiration in your grid cell averages. However, interception loss and evaporation should increase with a layer of mosses and lichens. If the water and energy balance is a topic in your paper, then please show results for all components, not only transpiration in Fig 12.**

*As explained above, the main focus of this article is to describe the implementation of boreal vegetation and only few key impacts, without a thorough analysis of the water, carbon and energy balances. However, we included additional diagnostics in the supplementary material, Fig. S5., with the main components of the water budget: evaporation (including interception), transpiration, runoff and drainage.*

**6) In this model version, two modifications affect soil temperature: snow depth and moss&lichen cover. First of all, the model version should be evaluated in terms of snow depth and soil temperature. For soil temperature, you can use GTN-P borehole data from Romanovsky et al. (2010) and Christiansen et al. (2010) available at PANGAEA, and maps of soil temperature and ALT even from your study region from Beer et al. (2013) at PANGAEA. I expect a cooling effect from mosses (Porada et al. 2016) due to higher insulation in summer, and a warming effect due to higher snow depth in areas of high shrub cover (still unclear to me at landscape scale as shrubs accumulate snow from lateral wind transport, so it is just relocated within the grid cell?). In Fig 13 both effects are combined. Is there a way to separate them? In Fig 13 it seems the model overestimates ALT and that is even higher in ORC16? In Fig 13b it seems all three grid cells show higher ALT (red) while in 13c one profile shows warmer temp (red) and the others show cooler temp? I generally suggest concentrating on soil temperature because ALT estimation from modelled temperature is not reliable.**

*We clearly understand the interest and your questions about soil temperature and water balance, key in the Artic to understand physical processes, e.g. the temporal dynamics of ALT and the evolution of permafrost. The Fig. 13 was made to illustrate small perspectives as a sample of the panel of potential impacts, but not as a comprehensive analysis. Given the current length of the paper, it was not possible to investigate these crucial questions in depth.*

*Additionally, to be exhaustive and perform proper evaluations of this insulating aspect, a factorial analysis would be needed, which was beyond the scope of this article. A dedicated study, with a different version of the ORCHIDEE model, ORCHIDEE-MICT, (including a description of the permafrost properties) has been conducted (Guimberteau et al. GMP,*

*submitted). In this context, we have chosen to illustrate only that the combined effect (summer and winter) is often more complex than expected with simplified formulations (although they remain important for understanding complex responses at global scales).*

*To represent the specific snow accumulation due to lateral wind transport and due to the lower snow compaction (itself due to branch support), the changes introduced (Section 2.3.2) are, as you suggest, just relocated within a grid cell. This is only applied in the case of the snow height used for the snow protection of shrubs (Equation 15).*

**7) Parameter estimation: Please show a priori and a posteriori parameter distributions in the appendix.**

*We added the corresponding supplementary: Table S2.*

**8) Please include a discussion section in which you interpret the results using literature in order to learn something. Parts of your summary section can be used if enhanced by literature. The conclusions and outlook section should be much reduced.**

*We acknowledge that the long "summary and conclusion" section (section 4) was maybe not the best choice to highlight the results of the study and replace them in the context of recent findings with similar models. We have chosen to follow the reviewer's advice and to split section 4 into a "discussion" section and a "conclusion" section (from p.22 l.17). The discussion now provides few interpretation of the results; however given the above-mentioned main objective of the paper (a model description), we do not provide a comprehensive interpretation of all carbon, water and energy related results. The conclusion has thus been reduced to the main key points of the paper, with an outlook of the next steps.*

**9) Several new methods are described but their importance, evaluation, and application is unclear: * Section 2.2.6: anoxic conditions are not simulated, soil organic matter dynamics are no topic of the paper. Please remove. Or was the intension to evaluate GPP and NEE at eddy covariance sites? * Why is shrub allometry important and why not only assume smaller trees? * Shrub-snow interactions are not evaluated or analyzed. What do we learn from these additional functions? * Effects on albedo: Has been albedo improved when comparing to satellite products?**

*We agree that there is probably a lack of evaluation of the new implementations described in the paper. The main reason comes from the need to keep the paper at a reasonable size and that a full evaluation including also a wider range of scientific applications has been left for a subsequent study. On the contrary we tried to represent the ecological complexity of vegetation, because biogeochemical and biophysical processes are interwoven.*

*Although we did not intend to evaluate the NEE at eddy covariance sites in this paper, we chose to include the modification linked to soil organic matter dynamics in order to provide a comprehensive model (for gross and net carbon fluxes), including the major processes that needed to be improved for subsequent biogeochemical applications.*

*Specifically for lowlands/peatlands, the maximum decomposition rate simulated with a maximum water content (i.e. in anoxic conditions) is not physically coherent and thus needed revision.*

*For shrubs, change in allometry (compared to trees) is the key process implemented for their representation: i) the initial tree allometry equation did not allow trees smaller than 10 meters, ii) this allometry impacts directly the mean and maximum values of biomass, which can be accumulated, iii) the height of the vegetation (and particularly the shrubs) is very important to take into account the snow temperature and protection (to maintain biomass in winter). The shrub-snow interaction is not precisely evaluated or analyzed as we believe the first priority is to evaluate whether the shrub biomass (i.e. including height, number of individuals,..) is realistically simulated.*

*The same concerns apply for the albedo, knowing that only the processes controlling the albedo of the snow were updated, and that the albedo of each new PFT has been kept to that of the original PFT (as a first approximation). Additional work is needed to fully characterize the albedo of the new PFTs and for NVPs its dependence to moisture conditions. This work is beyond the scope of the paper and we thus decided not to focus on a global evaluation of the albedo with existing satellite products.*

*In conclusion, it would have been orthogonal to the main objective of the paper to neglect key processes controlling the biogeochemical and biophysical functioning of the new boreal PFTs. But the evaluation and application of all of these aspects is impossible in one (already too long) article.*

**Minor issues:**

**Fig 10: not used in results but only in summary and that there also the fig does not support the sentence.**

*The Fig. 10 was use and directly mentioned in the result (Section 3.2., in the first submitted version from p.19 l.20 to p.19 l.34). However, to be more clear and concise we have decided to move this figure, as well as the figures 8 and 9, to the supplementary (Fig. S4).*

**$CO_2$ conductance in non-vascular plants depends strongly on its moisture and not on stomatal conductance. If that concept is not used here, then please discuss this limitation and related potential biases in detail.**

*We agree and it is for this reason that we have modified the constant value of the variable named "stomatal conductance" (Section 2.2.1., Eq. 1 and 2.) to reduce its dependence to active stomata and increase its dependence to moisture.*

**Page 16, line 35: I do not understand.**

*This was a description of the list of optimized parameters. As you suggested, it is now more explicit with the appendix (Table S2). In addition, these lines have now been moved at the end of Section 2.6.2.*

---

## Author Comment (AC1)

**Response to Referee #1**

*We thank R1 for this detailed review, especially for going through the equations, which has enabled us to significantly improve the description of the new process implementation in our article. We apologize for the erroneous formulations of several equations and have corrected them in the revised manuscript. We double-checked in the code that the lines of code correspond exactly to the revised formulation of equations. Enclosed please find a detailed explanation of the revisions we made based on R1's comments. For your convenience, comments are in bold and our response is in italic. Revisions we made in the manuscript are presented in italic with grey background.*

**This paper represents a great amount of work in model development, and in general it is well justified, well written, and the availability of such a model will contribute towards science both through using the improved model and informing other model developers. Therefore I recommend that it should be published in this journal, but with some clarifications and a bit of consolidation.**

**Firstly, the paper is rather long. I am not convinced that separating the analysis in figures 8-10 into different continents (Europe/Asia/America) is really relevant to the model developments here. Differences between the PFT's should still be visible in the aggregate results. Consolidating these would reduce the figures and you could remove some of the discussion of inter-continental differences from the text. These are interesting but the paper would benefit from being a bit shorter.**

*We are aware that the article is rather long: this is due to our wish to introduce together the two or three vegetation types needed to improve the current representation of Artic vegetation in the ORCHIDEE model. As the reviewer suggests, we removed the division by continent for figures 8-10 and the analysis associated (in Sect. 3.2 & 4.2), which was replaced by the Fig. 11. The old figures and analysis by continent is moved to the supplementary material (Figs. S1 to S3).*

**Throughout the manuscript you have used the word "summergreen", which I have never heard before and we always use "deciduous". I'm not sure summergreen in really a word in English and maybe you should used deciduous instead? Sorry if I'm wrong here.**

*In the model ORCHIDEE, the use of the word summergreen is required to compare the deciduous summergreen and raingreen (present only in tropical climates, as presented in Table 1). Considering that we are working only on boreal landscapes, it seems simpler, as suggested, to use the word "deciduous" in this article (p3. l.15-17, p5. l.29, p.20 l.8, p.23 l.26-27, Table 1. and Table 2.)*

**Specific comments**

**\* P1 Line 17 what you mean by "a larger phenological plasticity" isn't entirely clear to me. Maybe because I am not a specialist in vegetation but I think this will be read by other 'general' land surface modellers so could maybe be a bit clearer. Do you mean the phenology varies more in the season? Or more quickly over time?**

*"Phenological plasticity" means that the phenology of the plant can be shifted under hard climatic constraints, without causing its death. To be clearer, in the article we added a short description (in brackets p.1 l.17-18): "(i.e. adaptability and resilience to severe climatic constraints)".*

**\* P1 Lines 23-26. Please check all of these numbers for the percentage changes. I can't find them all in the main text or they don't seem to be consistent - for example, the change in roughness is quoted as 25% in the main text (page 20, line 33), but 41% in the abstract.**

*We have checked all numbers (value and %) present in this article. We have corrected the mistake (p.21 l.14 "decrease of 41% from 55°N" and we added in the main text: p.20 l.13-14 "For example, the NPP is lower by 31% north of 55°N", p.20 l.38 "+3.6% North of 55°N", p.21 l. 20 "(-33% from 55°N), as expected mainly …", p.21 l. 36 "(+11% with 140 $km^3.y^{-1}$ north of 55°N)".*

**\* P5 line 6 "coefficients a1 and a2" - should be "b1" instead of "a2" as it seems to be called b1 in the table. Furthermore, you said you chose values so that stomatal conductance would not depend strongly on VPD but then the multiplier of VPD (b1) takes a larger value for NVP's than for the original grasses so this seems a bit counter-intuitive. Could you add a bit more explanation here?**

*Indeed, as you have pointed out, the coefficient should be "$b_1$" instead of "$a_2$" (p.5 l.23). For the second comment, as you noted, the objective was to reduce the dependency of stomatal conductance to the humidity and $CO_2$ concentration, so to reduce the second term of eq. (1). The only adjustable constants are in eq. (2) (with the calculation of $f_{VPD}$): to reduce $f_{VPD}$ we had to increase the term "$1/(a_1-b_1.VPD)-1$", and so to reduce "$a_1-b_1.VPD$". Our choice to modify $b_1$ is based on the fact that $a_1$ is the same constant for all PFTs, when $b_1$ was already dependent on the vegetation type (trees, C3 grasses or C4 grasses). We propose here not to add more detail to the article, which is*

**\* Section 2.2.3: For the NVP's, when you have negative NPP you induce a biomass loss function. But presumably the negative NPP itself should also lead to a biomass loss. I am interested to know how this works - are these are somehow linked or are they two separate loss terms?**

*This is a very good point. In ORCHIDEE there is no explicit biomass loss when the NPP is negative. If NPP is negative, this means GPP < Ra (respiration) and this leads to a loss of biomass by the respiring tissues (to support Ra). But here, for NVPs we added a new explicit (and unlinked) loss term, to compensate for a reduced leaf biomass mortality (compared to the C3 grass PFT used as the starting point) due to the suppression of seasonal leaf fall and the increase of leaf longevity. Moreover this loss of biomass appears also if the NPP is null (not necessary negative).*

**\* Section 2.2.3 and Figure 1. Why did you reduce the turnover again after a certain amount of time? (ie why does the line on figure 1 decrease again after it reaches its maximum?) It would be helpful to provide some evidence from the literature or some more scientific justification here.**

*The aim of this turnover, presented in section 2.2.3 Eq. (3) and Fig. (1), is to represent the behaviour of NVPs in extreme conditions, such as snow cover or dryness, during a long period (more than 1 month). If the turnover was maintained at the maximum ($k_{lmax}$) when there is no NPP, rapidly most of the biomass would be removed and the plant would die. But that doesn't correspond to general observations of the presence of NVP biomass after snow removal, or after long very dry periods (with the desiccation process). To account for this resilience, we propose to reduce the biomass loss after a certain period of severe conditions. Note that there is still some biomass lost due to senescence. As suggested, we added a small description p.6 l.17-18: "After a maximum, the turnover decreases in order to represent the induced resistance and thus survival to extreme conditions, i.e. under snow cover in winter or under dryness".*

**\* P6 A few issues around equation 4 (which is labelled as 3 by the way!). Underneath the equation you wrote "b is the daily leaf biomass" but this is in units of gCm^(-2), which doesn't have any units of time, so it isn't 'daily'? Do you mean the value gets updated daily? I suggest removing the word 'daily' here. However, there should be some units of time in the turnover rate and I think these might actually be in lcoef, which you have given as no units, I think this should maybe have units of day^(-1) or similar? Difficult for me to tell from the information here but please check it. Another point about this equation, why does it only apply when LAI>LAI_max instead of LAI>LAI_lim? Using LAI_max means it will jump from zero when you reach LAI_max, whereas if you**

**from zero. Maybe this was a typo, but if not, can you explain why you do it? Thanks!**

*As you suggest, the confusion came more from a lack of information/description than from real mistakes. We thank the reviewer a lot for these comments. Here are all the changes we made:*

- *The label of equation 4 was changed (p6. l.34).*
- *The "daily" was removed (p6. l.35), because it stood for "updated daily" and that could be confusing.*
- *We added the unit of $l_{coef}$: $d^{-1}$ (p6. l.35 and table 2).*
- *There was some confusion between $LAI_{max}$ used for the photosynthesis and $LAI_{lim}$. So we checked the $LAI_{xxx}$ and changed the syntax when that was necessary (p6. l.32-35).*

**\* P8 Equation (10). This is quite a complicated equation and it would be really useful to see what the moisture function actually looks like. I suggest you add a plot of this. I looked in the paper that you referred to but it was not easy to immediately see it, and the moisture function for respiration is important so would be great to include the plot here.**

*Indeed, this equation is very complex. We followed your recommendation adding a new figure (Fig. 4) in order to have a better understanding of the new function and some text in p.9 l.12: "Equation (10) and Fig. 4 describes..."*

**\* P9 line 4/5 says that albedo and roughness were set the same as C3 grasses. I guess for NVP's the roughness could be quite different from grass? Could you add a comment on possible differences? Either here or in the discussion.**

*The roughness of NVPs can probably be considered to be of the same order of magnitude (compared to shrubs and trees), because they are both less than few tens of centimetres. However the albedo is very different, because their colour can vary widely especially depending the hydric status. We add in the discussion p.25 l.15-18 some precision about this issue: "However the albedo of the new boreal vegetation is still considered the same as that of the PFT they are derived from, although the colours of these PFT may vary substantially, with important impact on the albedo. In particular for NVPs (Porada et al., 2016) the colour may vary according to the relative humidity (Hamerlynck et al., 2000), an effect that is not easy to model globally."*

**\* P10 Equation 11a) The text says it's a logarithmic function, but this does not seem to be the case? Equation 11b) Bottom line of fraction should have D^(gamma) not D^2 Given these equations, I am not sure it makes sense to fix the crown area but still vary the biomass and height. This means that the allometric relations don't hold (for the case without dynamic vegetation), because the allometric relations**

**are basically the relationship between height and area (or diameter- but these are related), yet you are varying the height and not the area. Could you comment on this? Are you assuming that the number of individuals changes in order to keep the crown area fixed? If so, please make that clearer in the text.**

*Equation 11.a) it not expressed as a logarithmic function, but in order to describe the appearance of this function, we can consider that the usual function closest to eq. 11.a is the logarithmic function: starting from 0, increasing similarly to the logarithmic function and assuming a maximum ($H_{max}$). To be clearer, we propose to replace the "is" by "resembles to" (p.10 l.14).*
*We corrected equation 11.b. ("D^gamma" in place of "D2"), p.10 l.22.*
*The last part of this comment is about a fundamental choice of the developments performed in this article with the aim to obtain a realistic height of the vegetation to compute roughness, albedo or the height of shrub above the snow. The two strongest constraints were that i) without activating the dynamical vegetation (DGVM) module the total area of each PFT was fixed and ii) to be consistent, the equations with and without DGVM have to be the same. In order to have the vegetation height as a function of the biomass, we chose to implement a dynamical height depending on the biomass, following these equations (Eq. 11). Thus, as noticed by the reviewer, to account for vegetation height and diameter variations within a fixed area, the number of individuals has to vary. As a consequence, we can have only few tall shrubs or many short shrubs for a given area and biomass. To be clearer in the text, we added p10. l.18-19: "and the number of individuals is adapted in order to keep the crown area fixed (Eq. (11.c. & d.))".*

**\* P10 Section 2.3.2 In the introduction you said that shrubs accumulate more snow in winter than trees (p3 line 13), but in this section you seem to treat them both together. What is the reason for this?**

*The initial aim was to represent differences of snow accumulation on vegetation, not usually represented in ESMs. In this paper we started with the most significant difference between woody and non-woody species. In order to represent the differences between shrubs and trees, we would need to take into account precisely the spatial heterogeneity (vegetation coverage...), the phenology (evergreen and summergreen) and the wind effects. Given the complexity of the involved processes, it was beyond the scope of this already long paper and we thus focused only on the woody vs non-woody species difference. The sentence in the introduction is general and defines the ultimate objective.*

**\* P10 equation (13) I can guess what you are doing here - assuming**

**accumulate much snow, and with a lot of shrubs of course the snow will be the same as the grid box mean because they are covering the whole grid box. But what is the justification for peaking in the middle? Maybe with just a few shrubs they would still accumulate snow? Did you get this function from somewhere or did you come up with it yourself? Could you either (in the first case) add a reference or (in the second case) give a bit more explanation of the physical reasoning?**

*This comment is very constructive. As mentioned above, we did not find a simple and robust approach in the literature to take into account the differences of snow accumulation on vegetation. The best solution would have been to separate the snow accumulation (and the energy balance) by vegetation type, but this was not possible within the scope of the study. We thus chose a simplified approach, as explained by the reviewer. However, in light of your comment, we realize that we probably over-simplified the equation: indeed few shrubs should still accumulate snow. With this suggestion we could revise the threshold used in Eq. 13 and define a new equation:*

*"$1+4.f_v$" if $f_v < 0.2$ and "$2.-f_v$" if $f_v \geq 0.2$.*

*That may produce more realistic snow depth variations, with a peak of snow depth for high vegetation if its fractional cover is 0.2 instead of 0.5. However, given the small overall impact that is expected with such change and the difficulties to launch again the optimization and validation we choose to keep the initial formulation but to add a comment in the text p.11 l.16-17 "Note that this equation is a heuristic formulation discussed in section 4". In the discussion we added: "Equation (13), with a maximum snow depth obtained for a fraction of high vegetation of 0.5, probably underestimates the impact of shrubs on snow in the case of low shrubs cover. Having only few shrubs still leads to significant snow accumulation. We suggest further investigation of this sub-grid scale parameterization, possibly with the use of a similar equation but where the maximum snow depth on shrubs would be obtained for high vegetation cover around 0.1 to 0.3, instead of 0.5."*

**\* P11 Equation (15) I am not sure I agree about the form of this. Because you are integrating, the mortality rate (as a fraction of biomass) depends on the height of the shrub. Imagine your temperature is just constant with z, then the mortality rate will be proportional to (H-Hmin) and thus higher for a taller shrub - despite both being at the same temperature. Is this something you wanted to include in the model? If so, you should discuss it. If not, I would suggest you instead divide the RHS of the equation (15) for Mce by (H-Hmin).**

*We thank the reviewer for spotting the inconsistency. We indeed forgot to divide the RHS in equation 15 by the height. However, we do not divide RHS by*

*mortality is not applied below "$H_{min}$". If the temperature is constant with z that means the mortality is applied only on the fraction of the vegetation above $h_{min}$: ((H-Hmin)/H).*

**\* P12 first paragraph: I don't quite understand what f_v_max is. Do you prescribe a certain fraction of the grid cell to be occupied by a PFT but then it doesn't necessarily occupy that whole fraction? Please explain this term a bit more.**

*$F_{v\_max}$ is the maximum fraction of the grid cell occupied by each vegetation type (PFT), prescribed in the case of no dynamical vegetation. However, for grasses (and NVPs), which don't have woody parts, we consider that the real fraction of vegetation cover can differ from $F_{v\_max}$. The idea is to take into account, for roughness and albedo, the lack of leaves in winter. We use the Leaf Area Index (LAI) as a proxy for the vegetation cover, as usually done in global models, with an exponential decay. In order to improve the text, we added two sentences: p.12 l.22 "The fraction of vegetation (fv) is used" and p.12 l.25-26 "to take into account the variation of leaf cover (for example absent for grasses in winter)".*

**\* P12 equation (17) - you do the weighted average in terms of 'log's, I assume this is standard procedure from somewhere but I haven't see it before. Please add a reference.**

*Indeed, it is a standard simplified way of doing it, as detailed in "Vihma and Savijärvi, 1991" (p.12 l.31). The main principle follows from turbulence theory and the computation of the so-called drag coefficient that is a log function of the roughness length.*

**\* P13 L21-24 not sure what you mean by these things: - "survival or estabilishment limits" - limits in terms of what? Temperature? - "a cumulated degree-day threshold for the development" - maybe here you mean "..for the development of leaves"?**

*We agree that the terms that we used were inaccurate. We added p14. l.69 "temperature" and changed the word "development" p14. l.7 to "plant growth".*

**\* P14 line 1, talks about methods for wetlands, but surely not all of your sites are wetlands?**

*The published and unpublished data provided by Peregon et al. are more about lowlands. We take these data because we did not find any other data with total living biomass and productivity, on different sites, with multi-annual observations, and for the three new PFTs. Aware of this limit, we added an evaluation with biomass measurements from two other transects,*

*2011b; and previous reports since 2007). The evaluation of the model with these observations is supported by a new figure, Fig. 9, and associated comment:*

*P.17 l.14-19: "We further compare the simulated biomasses with two other Arctic transects. The first one is the North America Arctic Transect (NAAT). It is situated in a continental area, and includes eight field locations (70°N 149°W to 79°N 100°W) sampled from 2002 to 2006 (Walker et al., 2011b) chosen as representative of zonal conditions. The second, located in a marine-influenced area, is the Eurasian Arctic Transect (EAT). It includes six field locations (58 to 73°N, between 67 to 81°E) sampled from 2007 to 2010 (Walker et al., 2008, 2009a, 2009b, 2011a)."*

*P.19 l.1-15: "Carbon stock with two Arctic transect*

*To evaluate the modelled biomass in other Arctic sites (not used in the calibration step), including uplands and lowlands, Fig. 9 shows scatter plots of observed and simulated biomass along two transects: the NAAT (North America) and the EAT (Eurasia) Artic Transect. The NVPs and shrub biomasses are relatively well reproduced by the model (i.e. within the error bars). For both PFTs, the standard deviation of the observations includes the 1:1 line, but the observed biomasses are on average higher than the simulated biomasses. Simulated shrub biomasses are biased low for the NAAT transect but not for the EAT transect.*

*In contrast, the mean value of observed biomass for boreal C3 grasses (Fig. 9.c) is low compared to the simulated biomasses for both cases. For half of the sites the simulated low biomass is in accordance with the observations, but for the other half the values are much larger ($> 300\,gC.m^2$ whereas the observation do not exceed $54\,gC.m^2$). Despite the optimization with observations from western Siberia (Fig. 7; leading to a decrease of biomass compared to temperate C3 grasses) there is likely an overestimation of the biomass for boreal C3 grasses, probably associated with an overestimated productivity."*

*Walker et al, 2011a: Vegetation of zonal patterned-ground ecosystemsalong the North America Arctic bioclimate gradient. Applied Vegetation Science 14, 440–463. Doi: 10.1111/j.1654-109X.2011.01149.x*

*Walker et al, 2011. 2010 Expedition to Krenkel Station, Hayes Island, Franz Josef Land, Russia, Data Report, Alaska Geobotany Center, Institute of Arctic Biology, University of Alaska Fairbanks, Fairbanks, AK. 63 pp.*

**\* P14 line 32/33, it seems odd that the Arctic grasses are assigned to cold climates but then they all end up in the South! Have you checked this?**

*We have checked again the distribution of the vegetation, and we obtain the same result. That corresponds also to the "boreal trees" limit around 50 °N*

*and to mountainous regions. Note that our definition of the boreal region, based on Koppen Geiger climatic zones, has a relatively large extent given that we grouped several "continental cold climate" zones of the Koppen Geiger classification.*

**\* P15 line 7/8 What was the justification for these new distributions, especially with the grass fraction. Why did you include grass but not include any shrubs? Also a bit concerning that your percentages don't add up to 100%. What is the rest?**

*In the standard ORCHIDEE version, the sparse vegetation class (from the ESA map) was distributed into 25% of trees + shrubs, 35% of bare soil and the rest as grasses; the NVPs were not considered. As explained in section 2.5, the too small cover of NVPs in the satellite – derived product, led us to propose a new interpretation of the sparse vegetation class for boreal regions (based on other artic land cover maps), i.e. 45% of sparse vegetation class is considered as NVP cover. Thus we removed 15% of bare soil and 30% of grasses to represent NVPs (see Table S1). To be clearer in the text, we added p.15 l.34-35:* "The remaining fraction of sparse vegetation (25%) has not been modified and is considered as a mix of trees and shrubs."

**\* P16 : last sentence in section 2.6.1 talks about simulations and spin-up with no context (eg forcing data, soil characteristics?). I assume that the same simulation protocol as described in 2.6.2 is used for these simulations, and you extract the closest grid cells? But then the start of the simulation that it refers to at the end of Section 2.6.1 is not the same as described in Section 2.6.2. You need to more clearly explain what simulations are done/used for the parameter optimization.**

*Indeed, the lack of details could lead to confusion. We did not include enough explanation about the set up of the simulations for the optimisation, which is different than for the evaluation step. The biggest differences come from the spinup (as already explained) and the spatial scale (at 0.5° for the optimisation). We clarified that in the text by adding p.16 l.34-36 and p.17 l.1:* "The simulation for the optimisation was done with CRU-NCEP meteorological forcing (Wei et al., 2014; Viovy, 2015), at 0.5° resolution".

**\* Section 3.1 - the first 3 lines here are more like methods than results. Can you make this an extra (final) section in the methods perhaps?**

*,This is a good point. We have changed accordingly (p.17 l.23-30).*

**\* P17 line 23 How do you know the water stress in the model is too large? Could you show some evidence for this, or that it was seen in previous studies with ORCHIDEE?**

*The text was probably confusing, as we did not pretend that the water stress in the model is too large, in Arctic or elsewhere. We only have few grid points corresponding to the "forest-steppe", where the observations indicate a substantial vegetation development, when the model simulates a low development (low biomass). This forest-steppe ecosystem is situated at the foot of a mountain region (in the south), with less rainfall. So probably the local observation site has more soil water available for the plants than the large-scale (2°) mean soil water. To avoid misunderstanding (without much text increase), we add p.18 l.16-17: "due to too large a local water stress... at 2° resolution in a mountainous region".*

**\* P20 line 3/4 "too low LAI seems to be simulated in western Siberia" This looks more like the middle of Siberia to me?**

*We now write "in the central-west of Siberia" (p. 19 l.21) ?*

**\* P22 line 14 "plant resistance to water stress" - I thought you added something that made the NVP's recover more slowly from drought, and lose biomass, rather than resist the drought. Sorry if I missed the point here - do the other types of plants instead die in those circumstances? If so, could you clarify this?**

*This is complex, and given your comment, we realize that it was not clear enough in the manuscript.*

*For NVPs, first we removed the leaf fall and decreased the senescence. To partly compensate for that, we added a biomass loss when NPP≤0, but only during the first weeks in order to represent the cost linked to a resistance to extreme conditions. With this cost, the plant becomes more resistant and is able to survive during severe conditions. We made this more explicit (p.6 l.6-7 "extreme conditions introduced by lower leaf senescence and no leaf fall" and p.6 l.17-18 "After a maximum, the turnover decreases in order to represent the resistance induced and the survival, i.e. under snow cover in winter or under dryness").*

*Moreover, we reduced the maintenance respiration of NVPs in case of dryness (section 2.2.4) to represent the desiccation and the ability to resist efficiently to dryness. We added a reminder to these two processes p.22 l.34-35: "(through resistance to negative NPP (Sect. 2.2.3) and desiccation (Sect. 2.2.4))"*

**\* P22 line 32 "especially for NVP's" - Not sure about this. Aren't NVP's less nitrogen limited than other plants?**

*Indeed, the interest to introduce the NVPs is to be able to represent the vegetation in stressful condition. We wanted to insist that simulating in a realistic way any stress condition is important to estimate and model the*

NVP's" by p. 23 l.23-24: *"This is especially important for NVPs, which have an ecological advantage in these stressful conditions (such as poor nitrogen availability)".*

**\* P22 at the bottom of the page, you are talking about splitting shrubs into different types. It would be helpful to add in a comment about why it would be useful to do this? (What impact it might have?)**

*Such split would be useful to represent evergreen shrubs, which represent nearly half of shrubs cover in Arctic. Separating evergreen from deciduous shrubs can have important consequences, especially for the albedo in winter. We added in the text p.23 l.30-31: "such as evergreen phenology type, which represents more than 48% of shrubs North of 55°N according to the CCI product and Table S1"*

**\* P23 line 14/15, you are talking about how the seasonal cycle of NVP productivity differs from the vascular plants in the model, but there is no comment about whether these differences are realistic. You also mentioned earlier in the paper about 'representing the observed temporal dynamics of lichen and bryophyte biomass', but no reference to actual observations. It would be helpful to refer to some studies to discuss whether the behaviour of the model is realistic.**

*We agree that important ecological functionalities have to be justified by observations, using the literature. We thus have changed the text by adding p.24 l.8-10: "This behaviour corresponds to the observation that NVPs are, compared to vascular plants, most active during the shoulder seasons, due to less severe water stress and reduced competition for light (Williams and Flanagan, 1996; Campioli et al., 2009)"*

**\* P23 line 37/38, "the new PFTs are more sensitive to climate change than the original ones" - the plots do not seem to fully support this. The fractional changes are maybe larger with the new PFT's, but the 'old' PFT that you show on the plot (boreal broad- leaved trees) seems to have the largest absolute change and so potentially the biggest impact on the carbon cycle. I recommend modifying this discussion to account for this.**

*We agree that we have overstated the response of the new PFTs to climate change. We have to put into perspective the larger fractional changes, and the relative "absolute" contribution. To clarify that, we added p.24 l.33: "even if their overall contribution remains lower"*

**\* P25, Acknowledgements - I suggest you add more details of the projects, not just the acronyms i.e. full names and project numbers.**

*We add p.26 l.20-22: "The authors acknowledge financial support by the European Union Seventh Framework Programme (FP7/2007-2013) project PAGE21, under GA282700 as well as a French – Swedish program that has funded the first author's PhD, through the GAP project".*

**\* P38 Table 5. I think it is interesting that one of the calibrated parameters (b) was calibrated to zero. This appears to remove the acclimation behaviour from the photo-synthesis model. Could you add a comment about this in the text? Do you think it's because the air temperature never gets very warm so acclimation isn't necessary?**

*We thank the reviewer for this interesting suggestion. We therefore added p.23 l.4-8: "Note that for boreal C3 grasses the constant **b** of the entropy factor for the photosynthesis process (eq. (19) and eq. (20)) was optimized to zero (Tables 5 and S2), involving de facto the removal of seasonal temperature dependence of the photosynthesis process. That shows a limit of the Yin and Struik (2009) expression and could be due to the fact that the air temperature never gets warm enough to justify the need for acclimation."*

**Technical comments (In general the writing is good but I picked up some gram- mar/typos on the way through so will list these here.)**

*All of these technical comments were taken into account in the new version of the article.*

**\* P1 Line 24, "transpiration (+33%)" -> "transpiration (-33%)"**

**\* P2 Line 23/24, "is relatively simple and discretized on few" -> "has been relatively simple, with few"**

**\* P2 Line 26 "either trees or grasses PFTs." -> "either trees or grasses."**
**\* P2 line 27 "in the reality" -> "in reality"**

**\* P2 line 36 "interactions part" -> "interactions as part"**

**\* P3 line 4, I'm not sure about how you have referenced the CAVM, you have written "Mapping Team et al.", I wonder if it should just be "Mapping team" (and then the names listed are the members of the mapping team, not additional people?)**

**\* P3 line 7 "does not allow to" -> "does not allow it to"**

**\* P3 line 9 "mosses and lichens and shrubs" -> "mosses, lichens and shrubs"**

**\* P3 line 12 "more resistant for hydric" -> "more resistant to hydric" And "or for nitrogen limitation" -> "or to nitrogen limitation"**

* P3 line 15 "to warming whereas trees" -> "to warming, whereas trees"

* P4 line 16 "C3 grasses plants" -> "C3 grasses"

* P6 line 1 "cold temperatures" -> "cold temperature"

* P6 line 33 "(use in ORCHIDEE)" -> "(used in ORCHIDEE)"

* P7 line 13 "when NVP get desiccated." -> "when NVPs get desiccated."
* P7 line 30 "NVPs layer" -> "NVP layer"

* P8 line 24 "to define the control litter" -> "to control litter"

* P9 line 12 "processes as trees." -> "processes to trees."

* P9 line 22 "additional shrubs types" -> "additional shrub types"

* P11 line 2 "dynamically the vegetation distribution" -> "the vegetation distribution dynamically"

* P11 equation 16 Change 'else' to 'otherwise'

* P12 line 6 "there is no woody" -> "there are no woody"

* P12 line 27 "equation described previously" -> "equations described previously"

* P12 line 27 "as well as few" -> "as well as a few"

* P12 line 29 "Cold climates" -> "Cold climate"

* P12 line 34 "themselves function of" -> "themselves functions of"

* P13 line 12 "list of variable" -> "list of variables"

* P13 line 31 "observations located in" -> "observations are located in"

* P16 line 9 "number of iteration" -> "number of iterations"

* P18 line 12 "referred as" -> "referred to as"

* P20 line 25 "occur in early spring" -> "occurs in early spring"

* P20 line 27 "impact the albedo" -> "impacts the albedo"

* P20 line 34 "Contrariwise" -> "Conversely"

* P21 line 14 "5mmd-1" should be "0.5mmd-1" ?

* P21 line 20 "permanent frozen soil" -> "permanently frozen soil"

* P22 line 27 "implies to introduce" -> "implies introducing"

* P22 line 30 "availably" -> "availability"

* P23 line 19 "on the same time" -> "at the same time"

* P24 line 7 "in liason with" -> "in conjunction with"

* P24 line 13 "ecosystem occur" -> "ecosystems occur"

* P24 line 33 "permafrost extension" -> "permafrost extent"

* P24 line 33 "soil water dynamic" -> "soil water dynamics"

* P25 line 5 "and reach around" -> "and reaches around"

* P25 line 11 "reduce locally" -> "locally reduce"

* P25 line 12 "snow dynamic" -> "snow dynamics"

* Table 2 (df) "Maximum number of day for this extra turnover" -> "Maximum number of days..."

* Table 3 caption "values are choose" -> "values are chosen"